# Regulation of multispanning membrane protein topology via post-translational annealing

**Reid C Van Lehn, Bin Zhang, Thomas F Miller III\***

Division of Chemistry and Chemical Engineering, California Institute of Technology, Pasadena, United States

**Abstract** The canonical mechanism for multispanning membrane protein topogenesis suggests that protein topology is established during cotranslational membrane integration. However, this mechanism is inconsistent with the behavior of EmrE, a dual-topology protein for which the mutation of positively charged loop residues, even close to the C-terminus, leads to dramatic shifts in its topology. We use coarse-grained simulations to investigate the Sec-facilitated membrane integration of EmrE and its mutants on realistic biological timescales. This work reveals a mechanism for regulating membrane-protein topogenesis, in which initially misintegrated configurations of the proteins undergo post-translational annealing to reach fully integrated multispanning topologies. The energetic barriers associated with this post-translational annealing process enforce kinetic pathways that dictate the topology of the fully integrated proteins. The proposed mechanism agrees well with the experimentally observed features of EmrE topogenesis and provides a range of experimentally testable predictions regarding the effect of translocon mutations on membrane protein topogenesis.

**\*For correspondence:**
tfm@caltech.edu

**Competing interests:** The authors declare that no competing interests exist.

## Introduction

Integral membrane proteins (IMPs) are central to cellular functions that include signal transduction, transport across the cell membrane, and energy conversion. Performing these roles requires integration of the IMPs into the membrane with the correct topology (i.e., the correct orientation of the fully integrated IMP relative to the membrane). In most cases, membrane integration proceeds via the Sec translocon, a conserved protein-conducting channel located in the endoplasmic reticulum membrane in eukaryotes or in the plasma membrane in bacteria (*White and von Heijne, 2004*). During this process, the ribosome or other molecular motor docks to the cytoplasmic opening of the translocon, feeding the nascent protein into the translocon channel (*Shao and Hegde, 2011*); conformational changes in the lateral gate (LG) helices of the translocon then allow sufficiently hydrophobic segments of the nascent protein to integrate as transmembrane domains (TMD) (*Hessa et al., 2005*; *Egea and Stroud, 2010*; *Zhang and Miller, 2010*; *Gogala et al., 2014*). The orientation of a single TMD relative to the membrane is determined by factors that include the hydrophobicity of the TMD and the charge and length of the soluble loops that flank the TMD (*Goder and Spiess, 2001*, *2003*; *Devaraneni et al., 2011*). However, the extent to which these factors influence the topology of multispanning IMPs is less clear.

The conventional model of multispanning IMP topogenesis assumes that a single dominant topology is established via the successive integration of TMDs that thread back-and-forth across the membrane in alternating orientations (*Blobel, 1980*; *Wessels and Spiess, 1988*; *Sadlish et al., 2005*). In this cotranslational model, the dominant IMP topology is determined by the orientation of the N-terminal TMD and is primarily dictated by the features of that leading TMD (*Hartmann et al., 1989*;

**eLife digest** Proteins are long chains of smaller molecules called amino acids, and are built inside cells by a molecular machine called the ribosome. Many important proteins must be inserted into the membrane that surrounds each cell in order to carry out their role. As these proteins are being built by the ribosome, they thread their way into a membrane-spanning channel (called the translocon) from the inner side of the membrane. Short segments of these integral membrane proteins (called transmembrane domains) then become embedded in the membrane, while other parts of the protein remain on either side of the membrane.

For a membrane protein to work properly, the end of each of its transmembrane domains must be on the correct side of the membrane (i.e., the protein must obtain the correct 'topology'). The conventional model for this process suggests that topology is fixed when the first transmembrane domain of a protein is initially integrated into the membrane, while the ribosome is still building the protein. This model can explain most integral membrane proteins, which only have a single topology. However, it cannot explain the family of membrane proteins that have an almost equal chance of adopting one of two different topologies (so-called 'dual-topology proteins').

Van Lehn et al. have now used computer modeling to simulate how a bacterial protein called EmrE (which is a dual-topology protein) integrates into the membrane via the translocon. The results reveal that a few transmembrane domains in EmrE do not fully integrate into the membrane while the ribosome is building the protein. Instead, these transmembrane domains slowly integrate after the ribosome has finished its job.

These findings contradict the conventional model and suggest that some membrane proteins only become fully integrated after the protein-building process is complete. The next step in this work is to experimentally test predictions from the computer simulations.

*Borel and Simon, 1996*; *Dale et al., 2000*). However, the cotranslational model is challenged by dual-topology proteins, which exhibit both possible orientations of the fully integrated IMP with respect to the membrane in approximately 1:1 stoichiometry (*Rapp et al., 2006*, *2007*). The most thoroughly studied dual-topology protein is the bacterial multidrug transporter EmrE (*Chen et al., 2007*), which can be biased in favor of a single dominant topology by introducing positive charges to any of its soluble loops (*Rapp et al., 2006*, *2007*; *Seppälä et al., 2010*). The dominant topology of each EmrE mutant retains the loop with the additional positive charges in the cytoplasm (*Seppälä et al., 2010*), apparently satisfying the empirical trend known as the 'positive-inside' rule which notes that the combined charges of the cytoplasmic loops (i.e., K+R bias) of an IMP correlates with its dominant topology (*von Heijne, 1986*). Surprisingly, adding charges to even C-terminal loops can influence the dominant topology of EmrE, suggesting that such mutations have a long-range effect on the orientation of previously-translated TMDs. This finding is inconsistent with the cotranslational model and raises interesting questions about IMP topogenesis. At what point is IMP topology established with respect to ribosomal translation? Are TMD orientations locked-in during the period in which the nascent IMP is attached to the ribosome (i.e., cotranslationally) or do TMD orientations remain subject to change even upon completion of ribosomal translation (i.e., post-translationally)?

In this work, we simulate the topogenesis of EmrE and its mutants to address limitations in the cotranslational model of IMP topogenesis by understanding when IMP topology is established (co- or post-translationally) and how topology is regulated. We use a coarse-grained (CG) model that enables access to a timescale of minutes while retaining sufficient chemical accuracy to capture the forces that drive membrane integration (*Zhang and Miller, 2012a*). The distribution of topologies predicted by the simulations are in good agreement with previous experimental findings (*Rapp et al., 2007*; *Seppälä et al., 2010*). The simulation results show that TMDs in the dual-topology mutants do not completely integrate by the end of translation; instead, the slow post-translational flipping of loops across the membrane allows misintegrated TMDs to reorient and insert into the membrane. The fully integrated topology is determined by the position of the loop that undergoes flipping most slowly. This work elucidates the mechanism by which dual-topology protein topology is established, reconciles dominant protein topologies with the positive-inside rule, and predicts the role that the translocon plays in mediating multispanning IMP topogenesis. Other examples of post-translational

topological changes in diverse multispanning IMP systems suggest that this mechanism may have generality beyond EmrE (*Lu et al., 2000*; *Lambert and Prange, 2001*; *Kanki et al., 2002*; *Skach, 2009*; *Öjemalm et al., 2012*; *Bowie, 2013*; *Virkki et al., 2014*).

## Coarse-grained model

The cotranslational integration and topogenesis of EmrE and its mutants is simulated using a recently developed CG model (*Zhang and Miller, 2012a*), which we employ essentially unchanged from its original introduction. *Figure 1* illustrates the CG representation of a nascent protein and the protocol for simulating membrane integration. The ribosome, translocon, and nascent protein are all composed of CG beads. Each bead has a diameter of $\sigma = 0.8$ nm to represent approximately three amino-acid residues. This bead diameter is similar to the Kuhn length of polypeptides (*Staple et al., 2008*) so that the nascent protein can be treated as a freely jointed chain. The surrounding solvent and lipid bilayer are included implicitly, a technique that is used in other CG models of the translocon (*Rychkova and Warshel, 2013*). The time-evolution of nascent protein configurations is calculated using Brownian dynamics with a 100 ns timestep. The kinetics of the LG are modeled as stochastic transitions between a closed conformation, which prevents the nascent protein from exiting from the

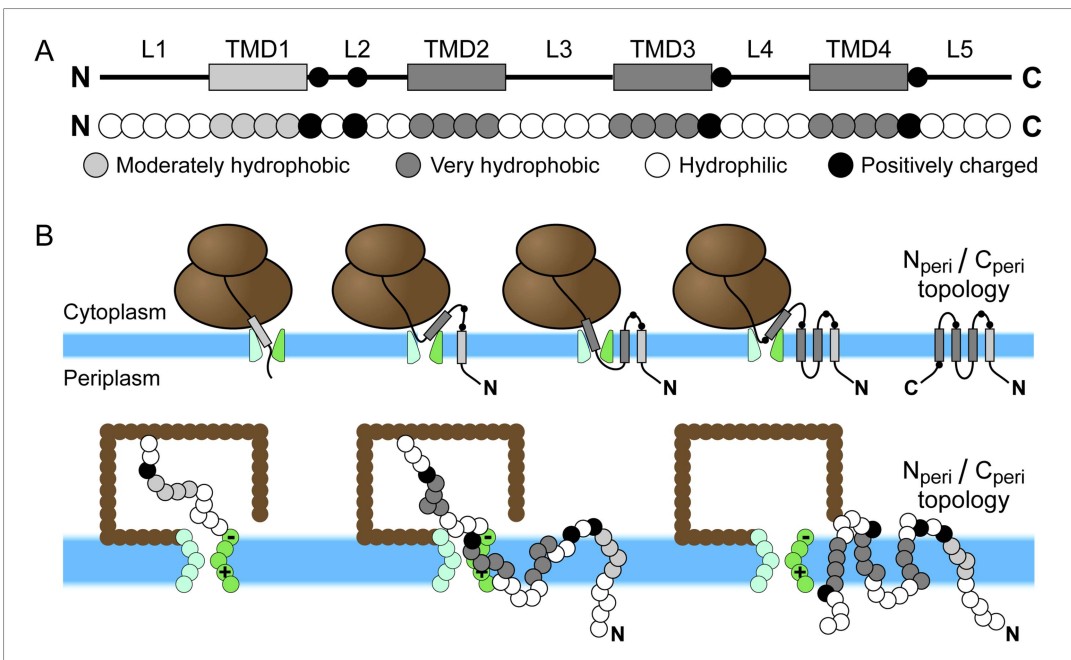

**Figure 1**. Schematic of Sec-mediated cotranslational integration of EmrE and corresponding simulation representation. (**A**) At top, an illustration of the structural motifs in EmrE, including indication of the charged residues in the soluble loops with black circles and the transmembrane domain (TMD)/loop numbering scheme that is employed in the text; below, the corresponding sequence of coarse-grained (CG) beads that represent the EmrE amino-acid sequence. TMDs and loops are assigned based on the hydropathy plot and consensus topology prediction shown in *Figure 1—figure supplement 1*. (**B**) At top, a schematic illustration of the sequential integration of TMDs to obtain a multispanning $N_{peri}/C_{peri}$ topology, in which both the N- and C-terminal loops are positioned in the periplasm, according to the cotranslational model; below, representative simulation snapshots of EmrE as the nascent protein grows during translation, integrates into the membrane, and exits the channel in the $N_{peri}/C_{peri}$ multispanning topology. The nascent protein is colored according to the legend at top, the ribosome is brown, and the translocon is green with translocon charges labeled explicitly.

The following figure supplements are available for figure 1:

**Figure supplement 1**. Hydropathy plot for EmrE.

**Figure supplement 2**. Simulation snapshot illustrating the initial configuration comprised of 9 CG beads.

channel interior to the membrane, and an open conformation, which removes the barrier to membrane entry. All bead positions are projected onto the plane that passes along the translocon channel axis between the helices forming the LG. This off-lattice 2D approximation reflects the cylindrical geometry of the channel and is inspired by previous models of biopolymer translocation through nanopores (*Huopaniemi et al., 2006*). Beads representing the ribosome enclosure and translocon are placed to approximate their structures (*Van den Berg et al., 2004*; *Frauenfeld et al., 2011*). Two negative charges are placed on a bead at the cytosolic end of the translocon LG, whereas two positive charges are placed on a bead at the periplasmic end of the translocon LG. This charge distribution reflects the position of conserved charged residues (*White and von Heijne, 2004*) near the translocon LG that have been previously shown to affect single-spanning protein topogenesis (*Goder et al., 2004*). Full details of the model are provided in Appendix 1.

The CG model is well-suited to simulating the kinetics of cotranslational IMP integration, a process that is challenging for atomistic models (*Zhang and Miller, 2010*; *Gumbart et al., 2011*; *Zhang and Miller, 2012b*; *Rychkova and Warshel, 2013*) due to the large system size (>100,000 atoms) and the long timescale (minutes) of translation. We note that the model does not include nascent protein secondary/tertiary structure, charged lipids, protein chaperones, or an electrostatic potential across the membrane. However, the model does include explicit LG/translation dynamics, electrostatic interactions with the translocon, water/bilayer transfer free energies, and a direct mapping between the nascent protein sequence and the CG representation. The model thus captures the major physicochemical features of the translocon-membrane system (*White and von Heijne, 2004*). Moreover, the model has been shown to accurately predict features of single-spanning IMP integration and topogenesis (*Zhang and Miller, 2012a*), including the sigmoidal dependence of stop-transfer efficiency on TMD hydrophobicity (*Hessa et al., 2005*), the inversion of signal-anchor orientation during translation (*Goder and Spiess, 2003*), and the effect of translation rates and sequence features on signal-anchor orientation (*Goder and Spiess, 2003*). In particular, the model has been shown (*Zhang and Miller, 2012a*) to correctly describe integration processes that are governed either by thermodynamics (*Hessa et al., 2005*) or kinetics (*Goder and Spiess, 2003*), and it has provided a means of understanding the competition between such effects. The model has also been shown to correctly predict the dominant topology for a three-TMD multispanning IMP with a strong positive-inside bias (*Zhang and Miller, 2012a*). The strong agreement between simulation and experimental results presented in this work further indicates that IMP topological determinants are captured at this CG resolution.

## EmrE protein

The EmrE amino-acid sequence includes four hydrophobic domains and five hydrophilic loops, according to both the hydropathy plot and consensus topology prediction shown in *Figure 1—figure supplement 1*. The hydropathy plot was calculated using the Wimley–White hydrophobicity scale (*Wimley et al., 1996*). The black line in the hydropathy plot indicates the water–octanol transfer free energy per residue and the overlaid red line shows a moving average using a 7-residue window. The consensus topology prediction was generated by the TOPCONS 1.0 server (*Bernsel et al., 2009*) and agrees with previous representations of EmrE structural elements (*Seppälä et al., 2010*). Shaded regions in the hydropathy plot indicate the predicted TMDs and loops.

In the CG model, each TMD is represented by four CG beads and each soluble loop is represented by five CG beads, as seen in *Figure 1A*. The CG beads assume one of four types as determined by the associated amino-acid residues in the nascent protein; these CG bead-types include V (moderately hydrophobic), L (very hydrophobic), Q (neutral-hydrophilic), and K (positively charged). Among these types, the CG beads vary with respect to their charge and their water/membrane transfer free energies (*Appendix table 1*). In the hydropathy profile, the N-terminal TMD (TMD1) is less hydrophobic than the other three TMDs, so its beads are assigned the V bead type. All other TMD beads are assigned the L bead type. Beads in each soluble loop are assigned to either the K or Q bead type, depending on the location of positive charges in the amino-acid sequence; positive charges are highlighted in red in the EmrE wild-type amino-acid sequence in *Figure 1—figure supplement 1*. Each K bead type is assigned a +2 charge, following previous work (*Zhang and Miller, 2012a*). Negative charges are excluded from the CG representation of EmrE, because EmrE exhibits a small number of such charges (*Figure 1—figure supplement 1*) and because the experimentally studied EmrE mutations focus only on the addition/removal of positively charged residues (*Seppälä et al., 2010*). Nonetheless, the effect of negatively charged residues in the CG simulation was explicitly

tested in *Figure 5—figure supplement 1* and was found to be minor. Similarly, the results of the simulations are robust with respect to changes in the modeling of TMD1 hydrophobicity (*Figure 5—figure supplement 1*) and loop length (*Figure 3—figure supplement 3*).

Using the CG model, we consider a series of EmrE mutants from *Rapp et al. (2007)* and *Seppälä et al. (2010)*. We include EmrE mutants with single charge mutations—K3, T28R, A52K, L85R, and R111—from *Seppälä et al. (2010)* and EmrE mutants with single dominant topologies—EmrE($N_{cyto}$) and EmrE($N_{peri}$)—from *Rapp et al. (2007)*. We also consider a series of mutants in which the protein has either zero positive charge or positive charges in only a single loop—nEmrE, nK3, nT28R[1], nT28R[2], nT28R, nA52K, nL85R, and nR111—from *Seppälä et al. (2010)*. This list includes all 16 of the EmrE and nEmrE mutants with single added charges studied experimentally by *Seppälä et al. (2010)*; mutants with added C-terminal His residues or an extra TMD are not considered. Finally, we include a 'cotranslationally-biased', or CB, mutant that has elongated, 10-bead hydrophilic loops and two positives charges in the first, third, and fifth loops to create a strong K+R bias that favors a $N_{cyto}/C_{cyto}$ topology (i.e., with both the N-terminal and C-terminal loops in the cytoplasm) according to the positive-inside rule (*von Heijne, 1986*; *Rapp et al., 2006*); this protein is expected to be strongly biased towards membrane integration via the cotranslational mechanism, providing a useful comparison with the other EmrE mutants. The CG representation of each mutant is listed in *Appendix table 2*; for each mutant, charge mutations are reflected by changing between Q-type and K-type beads at the appropriate point in the sequence. Despite its simplicity, we emphasize that the CG representation captures the major features of EmrE and its mutants, including the number of TMDs/loops and the distribution of charges.

## Simulation protocol

As illustrated in *Figure 1B*, the dynamics of the ribosome/nascent protein/translocon complex is directly simulated using the CG model. Each CG trajectory is initiated with a short nascent protein attached to the ribosome exit channel; as a function of time, the nascent protein grows in length (while remaining attached to the ribosome) until it completes translation and is released from the ribosome. The dynamics of the nascent protein continue to be simulated until the protein reaches a fully integrated topology.

Simulations are initialized from equilibrated configurations of the nascent protein, initially comprised of 9 CG beads, with the C-terminus attached to the ribosome exit channel (*Figure 1—figure supplement 2*). Translation is performed by adding a new CG bead to the C-terminus of the nascent protein and attaching it to the ribosome exit channel; the previous C-terminus is released from the exit channel. The simulation is then continued for 125 ms before the next bead is added, a simulation time which corresponds to a translation rate of 24 residues/s (*Bilgin et al., 1992*). At the end of translation, the C-terminus is released from the ribosome exit channel and simulations are continued until all beads in the TMDs are at least $4.5\sigma$ from the origin and integrated with either a $N_{cyto}/C_{cyto}$ or $N_{peri}/C_{peri}$ topology. The ribosome remains bound to the translocon for the duration of all simulations (*Potter and Nicchitta, 2002*; *Schaltetzky and Rapoport, 2006*). The distance threshold ensures that the final configuration of the protein has exited from both the ribosome and translocon channel.

The trajectory termination criteria are designed to examine the effects of the Sec-facilitated membrane integration process on EmrE topogenesis. Specifically, it is assumed that upon reaching configurations in which all of the TMDs are integrated into the membrane, the protein topology remains irreversibly fixed for all subsequent times; physical processes that may lead to this irreversibility include the dimerization of EmrE proteins to form functional channels in the membrane (*Lloris-Garcerá et al., 2012*) or the degradation of undimerized EmrE proteins prior to topological inversion (*Woodall et al., 2015*). Given the symmetry of the membrane-protein interactions in the absence of the translocon, if the CG trajectories were allowed to run for infinitely long times to reach full equilibration after diffusing away from the translocon, the relative probability of the $N_{cyto}/C_{cyto}$ and $N_{peri}/C_{peri}$ topologies would be equal, regardless of the protein sequence. The employed trajectory termination criteria thus isolate the role of the non-equilibrium integration process in determining IMP topology. Demonstration of the robustness of the reported results to the cutoff values employed in the trajectory termination criteria are provided in the *Robustness checks for the trajectory termination criteria* section of the 'Materials and methods'.

The integration and orientation of a TMD is interpreted from the positions of hydrophobic beads in each TMD and the third bead in each hydrophilic loop. The coordinate system is defined with the

x-axis perpendicular to the bilayer (*Figure 1—figure supplement 2*). The origin is placed at the center of the channel such that negative x-values indicate cytoplasmic positions. A TMD is considered integrated if $-2\sigma \leq x \leq 2\sigma$ for all four hydrophobic beads, corresponding to positions within the implicit bilayer, and if all y-positions are outside of the translocon interior. A loop is considered to be in the cytoplasm if the position of the reference bead satisfies $x < -\sigma$ and in the periplasm if $x > \sigma$. The $N_{cyto}/C_{cyto}$ topology is reached if the first, third, and fifth loops are positioned in the cytoplasm and the second and fourth loops are positioned in the periplasm. The $N_{peri}/C_{peri}$ topology has the opposite loop positions as shown in *Figure 1B*.

For each mutant, 250 independent trajectories are performed for a total of 4000 CG trajectories and nearly 6000 min of aggregate simulation time. Error bars measure the standard error between 2 blocks of 125 simulated trajectories. Complete system configurations are saved every 50 ms while loop positions and TMD orientations are saved every 1 ms.

## Results

### Simulations match experimental observations of topology

For all 16 of the EmrE and nEmrE mutants with single added charges studied by *Seppälä et al. (2010)*, *Figure 2* compares the experimentally observed IMP topologies with the prediction from the CG

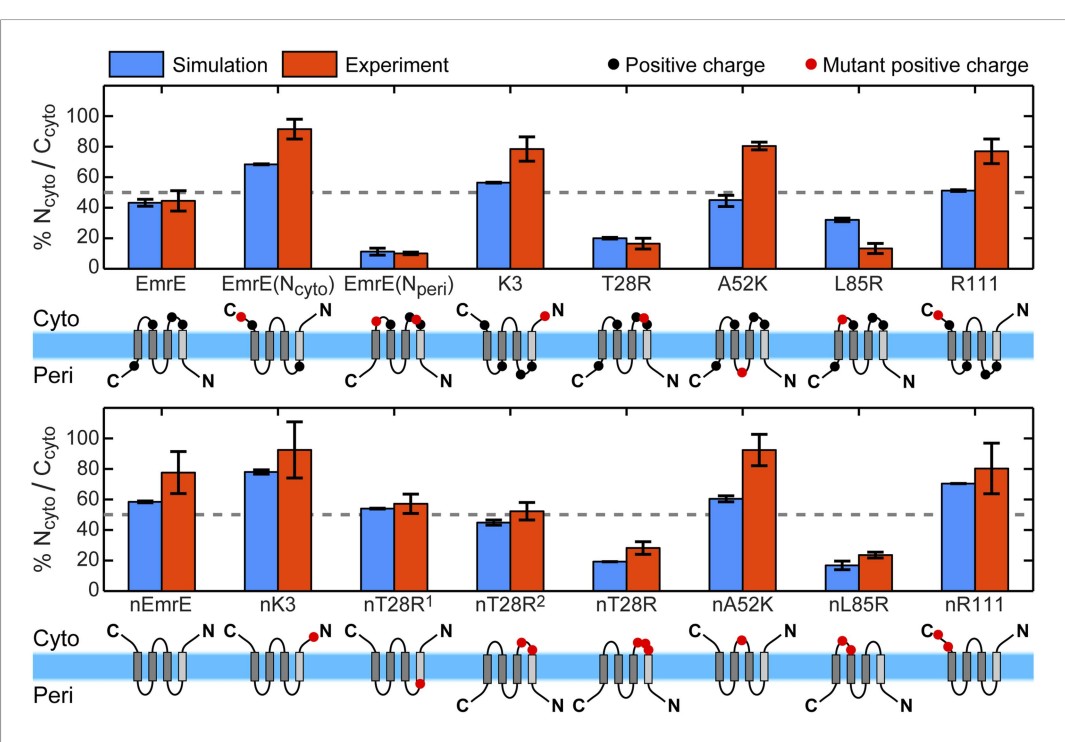

**Figure 2**. Topologies determined from simulations (blue) and compared to the experiments of *Seppälä et al. (2010)* (red), reporting the fraction of fully integrated integral membrane protein (IMP) configurations in the $N_{cyto}/C_{cyto}$ topology. Error bars indicate the standard error measured from independent blocks of simulations or taken from *Seppälä et al. (2010)*. The dominant topology for each mutant is indicated schematically with additional positive charges relative to EmrE (top) or nEmrE (bottom) drawn as red dots.

The following figure supplements are available for figure 2:

**Figure supplement 1**. Robustness of the distribution of topologies to the trajectory termination criteria.

**Figure supplement 2**. Correlation between the topologies determined from simulations (x-axis) and compared to the experiments of *Seppälä et al. (2010)* (y-axis), reporting the fraction of fully integrated IMP configurations in the $N_{cyto}/C_{cyto}$ topology.

model. Specifically, the figure compares the fraction of fully integrated proteins that adopt the $N_{cyto}/C_{cyto}$ topology, with the remainder in the $N_{peri}/C_{peri}$ topology. The top and bottom rows show variants of EmrE and nEmrE respectively. Each mutant differs only in the number and location of charges in the hydrophilic loops. A schematic of each mutant drawn with the dominant topology predicted from simulations is included; positive charges are indicated as filled-in circles with additional charges relative to EmrE (top row) or nEmrE (bottom row) highlighted in red. The topologies determined experimentally in *Seppälä et al. (2010)* are expressed as the fraction of $N_{cyto}/C_{cyto}$ topologies by dividing the cell activity of each protein coexpressed with the EmrE($N_{peri}$) mutant by the total growth of the protein coexpressed with either the EmrE($N_{peri}$) or EmrE($N_{cyto}$) mutant (*Seppälä et al., 2010*), as described in the *Experimental interpretation of EmrE topology* section of the 'Materials and methods'.

It is clear from *Figure 2* that the simulations are in excellent qualitative agreement with the experiments by correctly predicting the near 1:1 stoichiometry of wild-type EmrE and identifying the dominant topology for nearly all of the proteins considered. *Figure 2—figure supplement 2* illustrates that the distribution of topologies determined experimentally and the distribution of topologies measured from the simulations are linearly correlated (Pearson correlation coefficient, $r = 0.92$); points lying in the two shaded quadrants of the graph correspond to proteins for which the simulations and experiments predict consistent topologies. All mutants, with the exception of A52K, have the same dominant topology in the simulations as in the experiments within the statistical error. The agreement between simulations and experiments suggests that the CG model correctly reproduces the essential molecular features of topogenesis; in the following, we analyze the ensembles of CG trajectories that give rise to these computed IMP topologies.

## Dual-topology proteins exhibit slow post-translational integration

To investigate the molecular processes that govern the establishment of EmrE topology, we first examine the kinetics by which fully integrated topologies are reached. As a function of time, *Figure 3A* shows the fraction of CG trajectories in which the studied protein reaches a fully integrated topology for several EmrE mutants and the CB mutant. 0 s corresponds to the end of translation and negative values of time correspond to the period that precedes the end of ribosomal translation in which the nascent protein is still attached to the ribosome. Over 90% of the CB mutant trajectories reach the $N_{cyto}/C_{cyto}$ topology within 3 s following the completion of translation and thus rapidly integrate as expected for the cotranslational model (*Blobel, 1980*; *Wessels and Spiess, 1988*; *Sadlish et al., 2005*); mechanistic features of individual TMD integration steps are discussed in the *Cotranslational integration pathways* section of the 'Materials and methods'. In contrast, all variants of EmrE reach a fully integrated topology much more slowly, requiring hundreds of seconds for some CG trajectories to fully integrate (see also *Figure 3—figure supplement 2*).

The slow post-translational integration of the dual-topology EmrE mutants is due to the fact that a significant fraction of trajectories exhibit configurations in which some TMDs are not fully integrated at the end of translation. As a function of time, *Figure 3B* shows the fraction of CG trajectories in which each TMD is integrated for both the CB mutant (top) and EmrE (bottom). TMDs in the CB mutant integrate sequentially with near 100% efficiency during translation, which is consistent with the standard cotranslational model of topogenesis (c.f. *Figure 1*) and explains the rapid timescale for fully integrating into a multispanning topology shown in *Figure 3A*. In contrast, the TMDs of EmrE exhibit only partial integration, even at long times following the completion of translation. Snapshots of a typical misintegrated TMD in EmrE are shown in *Figure 3B*. Various experiments have indicated that such configurations with misintegrated TMDs arise due to frustration from charges placed in consecutive loops (*Gafvelin and von Heijne, 1994*), the strong orientational preference of a neighboring TMD (*Öjemalm et al., 2012*), or the weak stop-transfer efficiency of marginally hydrophobic TMDs (*Moss et al., 1998*). Consistent with these experimental observations, the simulations in *Figure 3B* find that the weakly hydrophobic TMD1 of EmrE integrates the least efficiently, followed by TMD4 which is flanked by two charged loops.

## The proposed mechanism
### Kinetic annealing of the end-of-translation ensemble
Analysis of the simulated CG trajectories reveals a straightforward molecular mechanism by which the multispanning topology of EmrE and its mutants is established. This mechanism, which we refer to as

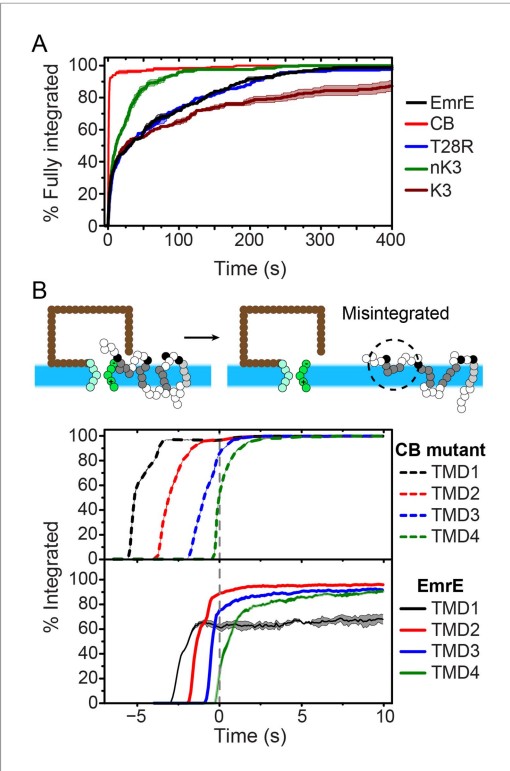

**Figure 3**. Kinetics of EmrE topogenesis. (**A**) Fraction of CG trajectories in which all TMDs are fully integrated in a multispanning topology, plotted as a function of time for several mutants. (**B**) Fraction of CG trajectories in which each TMD is integrated, plotted as a function of time for the cotranslationally-biased (CB) mutant (top) and EmrE (bottom). The snapshots show an example of a simulation in which TMD4 of EmrE does not integrate during translation. In both panels, 0 s corresponds to the end of translation and negative values of time correspond to the period that precedes the end of ribosomal translation.

The following figure supplements are available for figure 3:

**Figure supplement 1**. Pathways for the cotranslational integration of TMDs into the membrane.

**Figure supplement 2**. Simulation time necessary for 50%, 90%, and 95% of the CG trajectories to reach fully integrated topologies for each mutant.

**Figure supplement 3**. Effect of loop length on integration trajectories.

kinetic annealing of the end-of-translation (EOT) ensemble, is illustrated in *Figure 4* and involves two basic steps. In the first step, the cotranslational integration (or misintegration) of each TMD leads to an ensemble of IMP configurations associated with the time at which ribosomal translation completes; we call this set of configurations the EOT ensemble. In the second step of the proposed mechanism, configurations in the EOT ensemble anneal toward a fully integrated topology as a function of time as loops posttranslationally flip across the membrane. The rate at which the soluble loops undergo posttranslational flipping is a key determinant of the fully integrated topology. We will show that this mechanism explains the unexpected elements of EmrE topogenesis observed experimentally, including the topogenic effect of C-terminal mutations (*Seppälä et al., 2010*).

The first step of the proposed mechanism is presented in *Figure 4A* and *Figure 4B* in greater detail. As illustrated in *Figure 4A*, the EOT ensemble of each mutant is determined cotranslationally as TMDs exit the translocon. Differences in loop charges in the various mutants leads to variation in the corresponding EOT ensembles, because electrostatic interactions between highly-charged loops and the translocon favor their cytoplasmic retention (*Goder et al., 2004*; *Zhang and Miller, 2012a*). *Figure 4A* shows representative members of the EOT ensemble for the EmrE, T28R, and nR111 mutants with the most-charged loop of each mutant highlighted in red. The EOT ensemble is defined as the set of configurations visited by a given nascent protein within 1 s of simulation time following the termination of ribosomal translation. The schematics indicate how various TMDs integrate or misintegrate to give rise to heterogeneity in the EOT ensemble of configurations, while the loops with added charges preferentially obtain cytoplasmic positions. *Figure 4B* further quantifies the cytoplasmic bias of charged loops by showing the EOT ensemble averaged loop positions with respect to the membrane for all five loops in each mutant, expressed as the fraction of EOT configurations with a given loop in the cytoplasm. The increased cytoplasmic localization exhibited by the L2 and L5 loops in T28R and nR111

respectively highlights the effect of adding positive charges. Similarly, the reduced cytoplasmic retention of L2 and L4 in nR111 relative to EmrE is due to the removal of charges from these loops.

The second step of the proposed mechanism is presented in *Figure 4C* in greater detail. For each of the three mutants, the figure schematically illustrates the post-translational kinetics of two representative configurations from the EOT ensemble. Black horizontal arrows indicate how the flipping of soluble loops across the membrane leads to transitions between intermediate

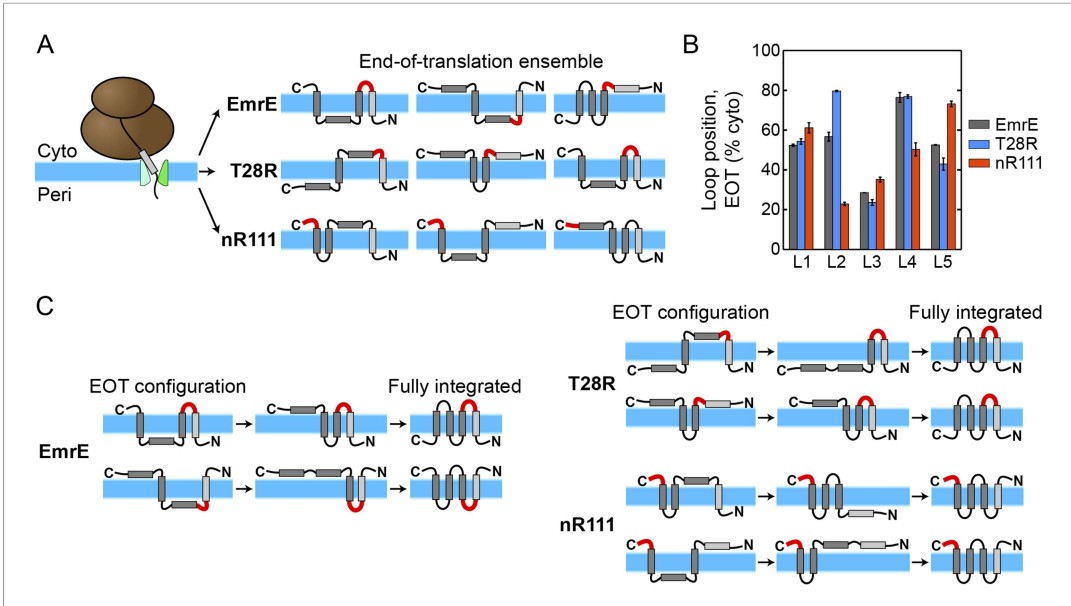

**Figure 4**. The proposed mechanism of kinetic annealing of the EOT ensemble. (**A**) Representative configurations visited by EmrE, T28R, and nR111 mutants within 1 s of simulation time following the end of ribosomal translation; this set of configurations for each mutant is called the EOT ensemble. The most-charged loop is indicated in red to highlight its position relative to the membrane. (**B**) The average loop positions from the EOT ensemble of each mutant is presented in terms of the fraction of configurations for which each loop occupies the cytoplasm. Adding charge to a loop biases towards a cytoplasmic position. (**C**) Post-translational annealing of representative EOT configurations for each mutant. Horizontal arrows indicate possible transitions as loops stochastically flip across the membrane to correct misintegrated TMDs. In each case, the position of the slowest-flipping loop (loop in red) determines the fully integrated topology by retaining its initial EOT position.

configurations. The most-charged loop is again highlighted in red. Each configuration post-translationally anneals toward a fully integrated topology as loops stochastically flip across the membrane to correct the misintegrated TMDs. The soluble loops undergo flipping at different rates, with charged loops flipping more slowly. The slowest-flipping loop thus determines the fully integrated topology that is most kinetically-accessible from a given configuration in the EOT ensemble, because the other loops will more rapidly flip. The EmrE examples (*Figure 4C*, left) demonstrate how the equal distribution of L2 positions with respect to the membrane in the EOT ensemble leads to two different fully integrated topologies, giving rise to the dual-topology behavior. The T28R examples (*Figure 4C*, right) show that increasing the charge of L2, thereby biasing its cytoplasmic localization in the EOT ensemble (*Figure 4B*), leads to a dominant $N_{peri}/C_{peri}$ topology. Finally, the nR111 examples illustrate how C-terminal charges can have a long-range topogenic effect by biasing the fully integrated proteins towards a dominant $N_{cyto}/C_{cyto}$ topology.

The proposed mechanism predicts that the final topological distribution of each EmrE mutant is determined by both the distribution of configurations in the EOT ensemble and the available post-translational kinetic pathways that lead to fully integrated protein topologies. In the following, we provide detailed analysis of the simulated CG trajectories to support these elements of the proposed mechanism.

## Charge mutations bias loop positions in the EOT ensemble

To investigate the first step of the proposed mechanism (*Figure 4A* and *Figure 4B*), we examine the degree to which changing the number of charges in a given soluble loop shifts the position of that loop in the EOT ensemble. *Figure 5* presents the average position with respect to the membrane of each individual loop of EmrE in the EOT ensemble (blue bars); as well as the corresponding average position of each loop in the mutant for which that loop includes an additional positive charge (red bars). Loop positions are expressed as the fraction of CG trajectories in which the loop is positioned in

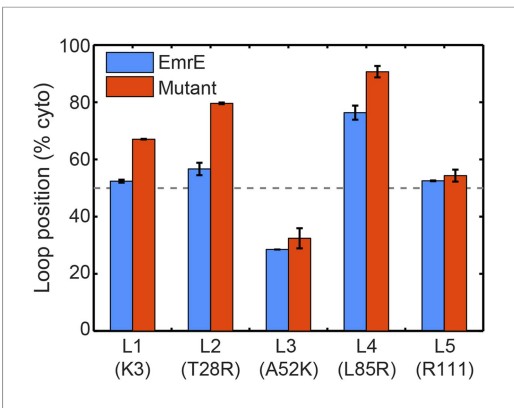

**Figure 5**. In blue, the average loop positions from the EOT ensemble of EmrE is presented in terms of the fraction of configurations for which each loop occupies the cytoplasm. In red, the corresponding average loop position from the EOT ensemble of EmrE mutants; for each loop, the presented result is for the charge mutation associated with that loop.

The following figure supplement is available for figure 5:

**Figure supplement 1**. The average loop positions from the EOT ensemble of six EmrE mutants is presented in terms of the fraction of configurations for which each loop occupies the cytoplasm.

the cytoplasm. In each case, the addition of positive charge to a soluble loop leads to an increase in its degree of cytosolic localization, as is consistent with previous simulations and experiments of single-spanning TMDs (*Goder et al., 2004*; *Zhang and Miller, 2012a*). These results support the first step of the proposed mechanism and show that interactions of the nascent protein with its translocon/ribosome/ membrane environment lead to significant shifts in the EOT ensemble of configurations.

## Rate of loop-flipping depends on charge mutations

To investigate the second step of the proposed mechanism (*Figure 4C*), we examine the molecular processes by which configurations in the EOT ensemble reach a fully integrated topology. The energetic cost for flipping a hydrophilic loop across the hydrophobic membrane increases with the hydrophilicity of the loop; the loop-flipping frequency observed during simulations is thus expected to decrease for loops with larger numbers of charges. *Figure 6* shows the computed loop-flipping frequencies for each loop in the EmrE mutants. In this analysis, loop-flipping events are determined by comparing loop positions with respect to the membrane in 1-ms time intervals, as described in the *Calculation of loop-flipping frequency* section of the 'Materials and methods'. The number of charges in each loop are marked with dots. As expected, highly-charged loops exhibit a decreased loop-flipping frequency. The figure also reveals that the terminal L1 and L5 loops have a lower loop-flipping frequency than the intermediate L2-4 loops. Loop-flipping events are not found to be strongly concerted, as two or more loops were observed to flip concurrently in only 0.015% of all 1-ms time intervals in which at least one loop-flipping event was observed. However, the loop-flipping frequency of a given loop is impacted by the orientation of its neighboring TMDs; on average, a loop with a single misintegrated neighboring TMD flips 1.5 times more frequently than the same loop with zero misintegrated neighboring TMDs, while a loop with two misintegrated TMDs flips 3.7 times more frequently than the same loop with zero misintegrated neighboring TMDs. Additional details on these calculations are presented in the *Calculation of loop-flipping frequency* section of the 'Materials and methods'.

The most important feature in *Figure 6* is the identification of a slowest-flipping loop for each mutant (red boxes). The slowest-flipping loop typically exhibits a loop-flipping frequency that is orders of magnitude slower than the other loops, although in four cases (K3, L85R, nEmrE, and nT28R[1]), two loops have slow loop-flipping frequencies that are within a factor of two. The variation in loop-flipping frequencies explains the difference in kinetics in *Figure 3A*, such that mutants with multiple slow-flipping loops (K3) reach a fully integrated topology more slowly than mutants with a single slowest-flipping loop (EmrE, T28R) or mutants largely devoid of charge (nK3). These results confirm that the loop-flipping frequency of a given loop depends strongly on the number of charges on that loop, indicating that charge mutations can impact the determination of the slowest-flipping loop.

## Position of slowest-flipping loop in EOT ensemble determines fully integrated topology

We now investigate the degree to which the position of the slowest-flipping loop in the EOT ensemble correlates with its position in the fully integrated topology. For the simulated CG trajectories, *Figure 7A* demonstrates strong correlation ($R^2 = 0.85$) between the position of the slowest-flipping loop in the EOT ensemble and the corresponding position in the fully integrated

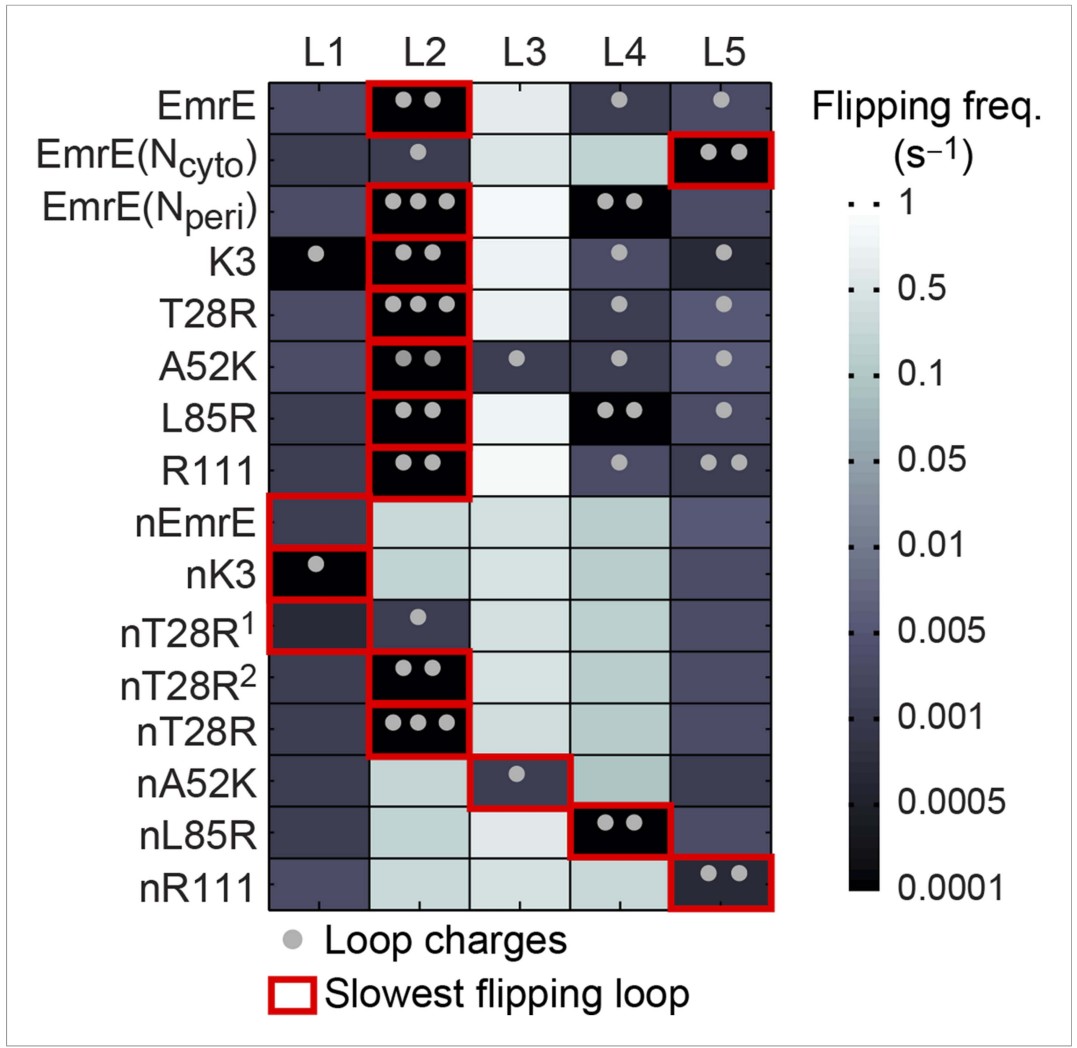

**Figure 6**. Loop-flipping frequencies computed from the CG trajectories. Darker squares correspond to less frequent loop-flipping events according to the logarithmic color map. Gray dots indicate the number of positively charged residues on each loop. For each mutant, the red box indicates the slowest-flipping loop.

configuration reached at the end of the CG trajectory. The results in *Figure 7A* indicate that the complexity of post-translational kinetics can be distilled to a much simpler picture in which the key parameter is the location of the slowest-flipping loop at the end of ribosomal translation. The fully integrated topology is almost completely determined at the time at which ribosomal translation ends, despite the fact that the kinetics of loop-flipping takes hundreds of seconds to complete.

In *Figure 7A*, the K3 and L85R mutants deviate most significantly from the plotted correlation between the EOT ensemble and the final topology; as seen in *Figure 6*, these two mutants exhibit a pair of slow loop-flipping frequencies rather than a single, well-separated slowest loop-flipping frequency. For a more detailed analysis of these special cases that involve a pair of slow loop-flipping frequencies, we direct the reader to the *Alternative definition of the slowest-flipping loop position for mutants with two slow-flipping loops* section of the 'Materials and methods' and the corresponding results in *Figure 7—figure supplement 1*. However, we emphasize that the close agreement between the results in *Figure 7A* and *Figure 7—figure supplement 1* indicate that our conclusions regarding the strong correlation between the EOT ensemble and the final topology are robust with respect to the details of the definition of the slowest-flipping loop.

The results in *Figures 2, 3* neglect the possibility that misintegrated proteins may be degraded prior to reaching a fully integrated topology. Several bacterial proteases that degrade membrane

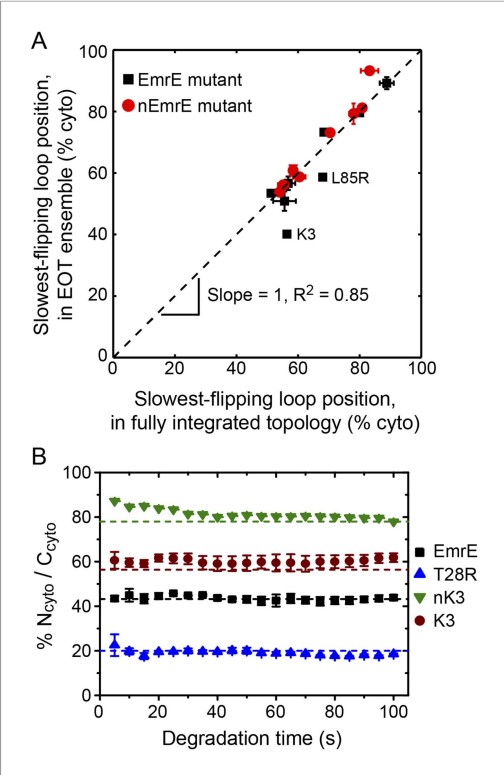

**Figure 7**. Relating configurations in the EOT ensemble to the fully integrated IMP topology. (**A**) Comparison of the average position of the slowest-flipping loop in the EOT ensemble to the average position of that same loop in the ensemble of fully integrated configurations. The average loop positions are presented in terms of the fraction of configurations for which each loop occupies the cytoplasm. The black dashed line indicates perfect correlation. (**B**) Fraction of $N_{cyto}/C_{cyto}$ topologies for the CG trajectories that have reached full integration, excluding all trajectories for which at least one TMD is misintegrated, plotted as a function of time following the end of translation. The dashed lines indicate the fraction of $N_{cyto}/C_{cyto}$ topologies for each mutant after all CG trajectories reach full integration, corresponding to the results from *Figure 2*.

The following figure supplement is available for figure 7:

**Figure supplement 1**. Comparison of the average position of the slowest-flipping loop in the EOT ensemble to the average position of that same loop in the ensemble of fully integrated configurations.

proteins have been characterized which provides insight into the approximate degradation time-scale (*Dalbey et al., 2011*). For example, FtsH is a membrane-embedded protease that degrades misassembled IMPs over timescales ranging from 2 min (for SecY) to 15 min (for YccA) in *Escherichia coli* (*Ito and Akiyama, 2005*), and even longer timescales for degradation have been observed in eukaryotic systems (*Buck and Skach, 2005*; *Feige and Hendershot, 2013*); very recently, FtsH was also shown to degrade undimerized EmrE on a sub-30 min timescale (*Woodall et al., 2015*). In comparison to the simulated trajectories (*Figure 3—figure supplement 2*), these degradation timescales are relatively slow, supporting the assumption that IMP integration and post-translational annealing reaches completion prior to significant degradation. Nonetheless, if degradation of EmrE occurs on faster timescales, it could potentially impact the reported topologies from the simulations. To investigate this effect, *Figure 7B* shows the relative fraction of $N_{cyto}/C_{cyto}$ and $N_{peri}/C_{peri}$ protein topologies for the CG trajectories that have reached fully integrated topologies as a function of time, excluding all trajectories for which at least one TMD is misintegrated. If it is assumed that fully integrated proteins are resistant to degradation (or that rapid dimerization following the full integration of EmrE protects the proteins from degradation [*Woodall et al., 2015*]), then each point in *Figure 7B* represents the distribution of topologies that would be observed if all misfolded proteins were uniformly degraded at the corresponding time. Data are shown for degradation times ranging from 5 s to 100 s following the end of translation; the dashed lines indicate the overall fraction of $N_{cyto}/C_{cyto}$ topologies for each mutant after all CG trajectories reach fully integrated topologies, corresponding to the results from *Figure 2*. *Figure 7B* shows that the distribution of topologies is nearly constant with respect to degradation time, preserving the correlation between the position of the slowest-flipping loop at the end of translation and in the fully integrated topology. These results suggest that the predicted distribution of protein topologies from simulation is relatively robust with respect to possible degradation processes that occur on the same timescale as post-translational annealing.

## Discussion

The results of our CG simulations support a mechanism for multispanning IMP topogenesis in which an ensemble of configurations with misintegrated TMDs undergo kinetically-controlled TMD reorientations to reach a fully integrated topology. Introducing charge mutations to the soluble loops of a multispanning IMP leads to shifts in both the distribution of loop positions in the EOT ensemble (*Figure 5*) and changes in the kinetics of loop-flipping events that lead to the fully integrated

topologies (*Figure 6*). The combination of these effects is found to govern the observed distribution of fully integrated topologies in the CG simulations (*Figure 7*). This proposed mechanism explains the experimental finding that adding charges to any of the soluble loops of EmrE, even a loop near the C-terminus, affects the observed topology (*Seppälä et al., 2010*). The proposed mechanism also agrees with recent experiments that find EmrE to undergo partial topological rearrangements that correspond to the loop-flipping events described here (*Woodall et al., 2015*). Furthermore, the mechanism can explain deviations from the positive-inside rule if the position of the slowest-flipping loop in the EOT ensemble enforces a topology in which the majority of the positive charges are in periplasmic loops, as seen for the K3 mutant (*Figure 2*).

In addition to explaining existing experimental data for the topogenesis of the EmrE mutants, the proposed mechanism yields a number of new and experimentally testable predictions. A simple overarching prediction of the mechanism is that changes to the ribosome or translocon that affect the EOT ensemble may lead to significant shifts in topology. *Figure 8* shows the average position of the slowest-flipping loop in the EOT ensemble after slowing translation from 24 residues/s to 6 residues/s to model the addition of cycloheximide (*Goder and Spiess, 2003*), removing the periplasmic positive charge from the channel, or removing the cytoplasmic negative charge from the channel (*Goder et al., 2004*). For single-spanning IMPs, the rate of translation and the removal of translocon charges were previously found to significantly affect TMD orientation in both simulations and experiments (*Goder and Spiess, 2003*; *Goder et al., 2004*; *Zhang and Miller, 2012a*). We find that slowing translation has a minimal effect on the mutants studied here, and *Figure 8—figure supplement 1* confirms this finding for other translation rates. Given that these EOT loop positions are unchanged, and given that the post-translational dynamics is unaffected by the ribosomal translation rate, these results suggest that changing translation rate will not affect the final distribution of fully integrated topologies. In contrast, *Figure 8* shows that removing either the cytoplasmic or periplasmic charge on the translocon significantly decreases the cytoplasmic retention of the slowest-flipping loops by increasing the periplasmic accessibility of highly charged loops. Most notably, it is found that for two of the EmrE mutants (indicated in dashed boxes) the translocon charge mutations dramatically shift the slowest-flipping loop position in the EOT ensemble from being primarily cytosolic to being primarily periplasmic, suggesting that the dominant topology for these EmrE mutants will be similarly reversed by the translocon charge mutations. These changes in IMP topology due to channel mutations are experimentally testable predictions of the proposed mechanism.

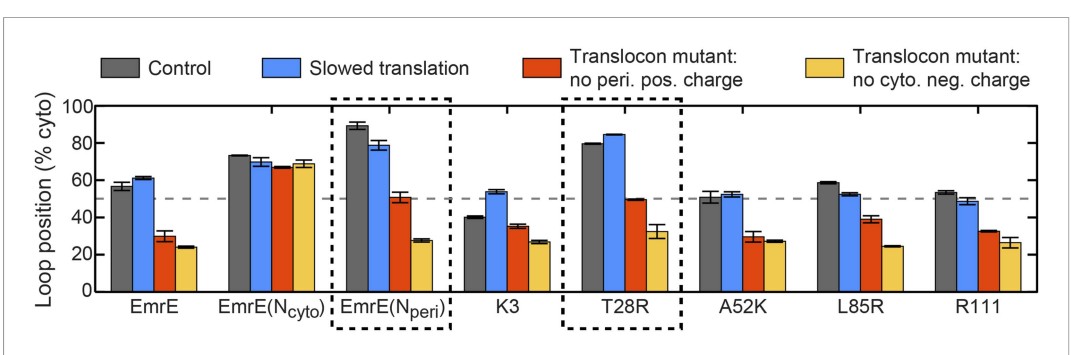

**Figure 8.** Predictions from the CG simulations in terms of changing the rate of ribosomal translation and introducing translocon mutations. In each case, the figure presents the average position of the slowest-flipping loop in the EOT ensemble. The average loop position corresponds to the fraction of configurations for which that loop occupies the cytoplasm. The first column (dark gray) shows loop positions from the control simulation protocol. The remaining three columns show loop positions associated with the fourfold slowing of ribosomal translation (blue), associated with removal of the positive charge on the periplasmic opening of the translocon (red), and associated with removal of the negative charge on the cytoplasmic opening of the translocon (orange). The dashed boxes emphasize EmrE mutants that are predicted to exhibit dramatic inversions of the dominant topology upon translocon mutations.

The following figure supplement is available for figure 8:

**Figure supplement 1**. Effect of changing the rate of ribosomal translation on loop positions.

A notable aspect of the CG model is the absence of asymmetric features in the membrane or environment that favor either the $N_{cyto}/C_{cyto}$ or $N_{peri}/C_{peri}$ topology under equilibrium thermodynamic conditions, such as the electrostatic potential across the inner membrane of *E. coli* or an asymmetric distribution of charged lipids (*Bogdanov et al., 2008*; *Vitrac et al., 2013*; *Bogdanov et al., 2014*). In the CG model, neglecting interactions with the Sec translocon, both the $N_{cyto}/C_{cyto}$ or $N_{peri}/C_{peri}$ topologies are energetically equivalent and would be observed with equal probability if simulations were continued for an infinitely long time. The prediction of a dominant topology by the CG model arises from the initial distribution of configurations in the EOT ensemble (due to interactions of the nascent protein with the translocon complex) and from the available kinetic pathways that allow the configurations in the EOT ensemble to reach fully integrated topologies. We note that the changes in topology predicted in *Figure 8* would be unexpected from a model in which the dominant topology of an IMP is determined by thermodynamic equilibrium, since the equilibrium distribution of the protein topologies would be unaffected by transient interactions with the translocon or ribosome during initial membrane integration.

We further note that direct comparison of the experimental and simulation timescales for the kinetic annealing of misintegrated TMDs is limited by both the accuracy of the CG model as well as the neglect of external chaperone proteins, such as TRAP, TRAM, or other members of the Sec complex (*Sommer et al., 2013*; *Zhu et al., 2013*; *Aviram and Schuldiner, 2014*; *Jung et al., 2014*), that may catalyze loop-flipping. However, since the topological predictions of the proposed mechanism are primarily sensitive to which soluble loop flips most slowly—as opposed to the actual timescale of loop-flipping—we expect that the presented conclusions are relatively robust with respect to these effects. This robustness is directly illustrated in *Figure 7B*, which shows that the relative fraction of proteins that reach each fully integrated topology is nearly constant as a function of time.

## Conclusions

In this work, we utilize a recently developed CG computational approach (*Zhang and Miller, 2012a*) that enables the direct simulation of Sec-facilitated membrane integration of proteins on biological timescales to investigate the topogenesis of the dual-topology EmrE protein and its mutants. In addition to demonstrating excellent agreement with the experimentally observed topologies of EmrE and its mutants (*Seppälä et al., 2010*), the simulations reveal a novel mechanism for the regulation of topogenesis in multi-spanning membrane proteins, in which initially misintegrated configurations of the proteins undergo post-translational annealing to reach final, fully integrated topologies. The energetic barriers associated with this post-translational annealing process enforce kinetic pathways that dictate the topology of the fully integrated proteins. The inclusion of charged residues on the soluble loops of the IMP can lead to significant changes in the distribution of fully integrated topologies by both altering the ensemble of protein configurations at the end of ribosomal translation, as well as by altering the available kinetic pathways that lead to fully integrated topologies.

This analysis leads to a number of experimentally testable predictions regarding IMP topogenesis. In particular, the results of *Figure 8* predict that the mutation of charged residues near the cytoplasmic or periplasmic openings of the translocon channel will lead to significant shifts in the observed topology of several EmrE mutants. More generally, we note that any effect of channel mutations on the fully integrated IMP topology would indicate that kinetic effects during translation influence topogenesis, as suggested by the proposed mechanism. Additionally, we predict that the introduction of IMP mutations that significantly alter the EOT ensemble with respect to the cytosolic localization of the slowest-flipping soluble loop, either by introducing charge mutations or by changing TMD hydrophobicity, will influence the multispanning IMP topology.

Although the current manuscript primarily focuses on the mechanism of topogenesis in the dual-topology EmrE protein, the mechanism and simulation analysis presented here has broader implications for topogenesis in other multispanning IMPs. For EmrE and its mutants, we find that a significant fraction of the IMP configurations are misintegrated upon completion of ribosomal translation and undergo subsequent post-translational annealing to reach fully integrated topologies. In contrast, a CB mutant exhibits an essentially fully integrated ensemble of configurations at the time that ribosomal translation completes. For other IMPs, a combination of these behaviors may well be expected (*Lu et al., 2000*; *Lambert and Prange, 2001*; *Kanki et al., 2002*; *Skach, 2009*; *Öjemalm et al., 2012*; *Bowie, 2013*; *Virkki et al., 2014*), with some fraction of the nascent protein

configurations reaching fully integrated topologies at the completion of ribosomal translation and some fraction reaching misintegrated configurations that subsequently undergo post-translational annealing. Indeed, the importance of chaperone proteins such as YidC or Sec62 that post-translationally rescue misintegrated TMDs (*Sommer et al., 2013*; *Zhu et al., 2013*; *Aviram and Schuldiner, 2014*; *Jung et al., 2014*) may be connected to this necessary process of annealing initially misintegrated IMP configurations towards fully integrated topologies. The emerging understanding of the role of the Sec translocon in regulating IMP topogenesis, as well as advances in the methodologies for probing and modifying interactions between the nascent protein and the translocon complex, hold intriguing possibilities for the prediction and control of protein folding in cellular environments.

## Materials and methods

### Calculation of loop-flipping frequency

Loop-flipping frequencies are calculated by monitoring loop positions with respect to the membrane in 1-ms time intervals. A loop-flipping event is counted if a given loop switches from a cytoplasmic to periplasmic (or vice versa) position according to the definitions in the main text and if the *y*-position of the reference bead in the loop (defined in the main text) is at least $4.5\sigma$ after the flip (to exclude counting the rapid flipping of loops within the translocon channel). The loop-flipping frequencies presented in *Figure 6* are obtained by averaging the frequency of loop-flipping events in each trajectory. Sufficiently infrequent loop flips will not occur in every trajectory, but such events are observed in the combined ensemble of trajectories.

To examine the effect of neighboring TMDs on the loop-flipping frequency, we separately calculate the loop-flipping frequency of each loop for configurations in which zero, one, or two of the neighboring TMDs is misintegrated (discussed in the *Rate of loop-flipping depends on charge mutations* section of the 'Results').

### Experimental interpretation of EmrE topology

In *Seppälä et al. (2010)*, the dominant topologies of EmrE mutants are determined by measuring the growth of *E. coli* cells in the presence of ethidium bromide (EtBr). EtBr is toxic to *E. coli*, but antiparallel EmrE dimers, in which the two monomers forming the dimer have opposite topologies, confer drug resistance. EmrE dimerization can also be suppressed by including an E14D mutation. The topology of an EmrE variant with the E14D mutation can thus be inferred by coexpressing the mutant with another EmrE variant of known topology, as any resulting drug resistance (and cell growth) can be attributed to the formation of antiparallel dimers. To enable a direct comparison between the topologies measured from simulations and the experimental results, we convert the experimentally-measured cell activities from *Seppälä et al. (2010)* to the fraction of $N_{cyto}/C_{cyto}$ topologies by assuming a linear relationship between cell growth and the population of antiparallel EmrE dimers. The fraction of $N_{cyto}/C_{cyto}$ topologies is calculated as

$$f\left(N_{cyto}/C_{cyto}\right) = \frac{A\left(N_{peri}\right)}{A\left(N_{cyto}\right) + A\left(N_{peri}\right)}, \tag{1}$$

where $A(N_{cyto})$ and $A(N_{peri})$ are the experimentally-measured cell activities for cells coexpressing the EmrE($N_{cyto}$) and EmrE($N_{peri}$) mutants, respectively. Greater cell growth in the presence of the EmrE($N_{peri}$) mutant, which exhibits a single dominant $N_{peri}/C_{peri}$ topology, indicates that the mutant of interest exhibits a larger fraction of the opposite $N_{cyto}/C_{cyto}$ topology, and vice versa for growth in the presence of the EmrE($N_{cyto}$) mutant. Experimental values for the activities of the EmrE and nEmrE mutants are taken from Figure 2 and Figure S1 of *Seppälä et al. (2010)*, respectively; these values are used to compute the fraction of $N_{cyto}/C_{cyto}$ topologies reported in *Figure 2* of the current paper. Values for the activities of the ($N_{out}$(E14D) + $N_{in}$) and the $N_{out}$ constructs from Figure 1 of *Seppälä et al. (2010)* are used to approximate the topology of the EmrE($N_{peri}$) mutant, while the activities of the ($N_{in}$(E14D) + $N_{out}$) and the $N_{in}$ constructs from the same figure are used to approximate the topology of the EmrE($N_{cyto}$) mutant. Error bars are approximated via standard error propagation techniques based on *Equation 1*.

## Alternative definition of the slowest-flipping loop position for mutants with two slow-flipping loops

In *Figure 7A*, the average position of the slowest-flipping loop relative to the membrane in the EOT ensemble is compared with the average position of that same loop in the ensemble of fully integrated configurations at the end of the CG trajectories. Four mutants (K3, L85R, nEmrE, and nT28R[1]), however, have two slow-flipping loops with similar loop-flipping frequencies (*Figure 6*), and two of these mutants (K3 and L85R) deviate most significantly in terms of the correlation in *Figure 7A*.

To better understand the effect of multiple slow-flipping loops on the correlation between the EOT ensemble and the final topology, the current section provides additional analysis in which a more sophisticated definition of the 'slowest-flipping loop' is employed for the four mutants that exhibit a pair of slow-flipping loops. Below, we present this alternative definition, which leads to a slightly better correlation between the EOT ensemble and the ensemble of fully integrated configurations, as plotted in *Figure 7—figure supplement 1*.

The alternative definition of the slowest-flipping loop for mutants with two slow-flipping loops is given by $\phi_{\text{EOT}}$ and $\phi_{\text{FI}}$, which report on the average position of the two slow-flipping loops in the EOT ensemble and in the ensemble of fully integrated configurations, respectively.

The quantity $\phi_{\text{FI}}$ reports on the average position of the two slow-flipping loops in the ensemble of fully integrated configurations at the end of the CG trajectories. For the L85R and nEmrE mutants, the two slow-flipping loops (L2/L4 and L1/L5, respectively) reach positions on the same side of the membrane in either fully integrated topology; for these mutants, $\phi_{\text{FI}}$ is defined as the fraction of fully integrated configurations for which both slow-flipping loops are positioned in the cytoplasm. For the K3 and nT28R[1] mutants, the two slow-flipping loops (L1 and L2) reach positions on opposite sides of the membrane in either fully integrated topology; for these mutants, $\phi_{\text{FI}}$ is defined as the fraction of fully integrated configurations for which L1 is positioned in the cytoplasm and L2 is positioned in the periplasm. For the nEmrE, K3, and nT28R[1] mutants, $\phi_{\text{FI}}$ is equivalent to the fraction of CG trajectories that reach the fully integrated $N_{\text{cyto}}/C_{\text{cyto}}$ topology, whereas for the L85R mutant, $\phi_{\text{FI}}$ is equivalent to the fraction of CG trajectories that reach the fully integrated $N_{\text{peri}}/C_{\text{peri}}$ topology.

The quantity $\phi_{\text{EOT}}$ reports on the average position of the two slow-flipping loops in the EOT ensemble. For each mutant, $\phi_{\text{EOT}}$ is defined as

$$\phi_{\text{EOT}}^{\text{L85R}} = 0.5\left[f(\text{cyto})_{\text{L2}} + f(\text{cyto})_{\text{L4}}\right]$$
$$\phi_{\text{EOT}}^{\text{nEmrE}} = 0.5\left[f(\text{cyto})_{\text{L1}} + f(\text{cyto})_{\text{L5}}\right]$$
$$\phi_{\text{EOT}}^{\text{K3}} = 0.5\left[1 + f(\text{cyto})_{\text{L1}} - f(\text{cyto})_{\text{L2}}\right]$$
$$\phi_{\text{EOT}}^{\text{nT28R}^1} = 0.5\left[1 + f(\text{cyto})_{\text{L1}} - f(\text{cyto})_{\text{L2}}\right], \tag{2}$$

where $f(\text{cyto})_{Li}$ is the fraction of configurations in the EOT ensemble for which loop L$i$ is in the cytoplasm. As for the previous definition of $\phi_{\text{FI}}$, this definition accounts for the fact that the two slow-flipping loops of the L85R and nEmrE mutants reach the same side of the membrane in the fully integrated topologies, while the two slow-flipping loops of the K3 and nT28R[1] mutants reach opposite sides of the membrane in the fully integrated topologies. The definition in *Equation 2* additionally assumes that the post-translational annealing of misintegrated configurations in the EOT ensemble is equally rate-limited by the two slow-flipping loops.

Using these alternative definitions for the position of the slowest-flipping loop (i.e., $\phi_{\text{FI}}$ and $\phi_{\text{EOT}}$), *Figure 7—figure supplement 1* compares the average position of the slowest-flipping loop in the EOT ensemble to the average position of that same loop in the ensemble of fully integrated configurations. Having more carefully accounted for the effect of both slow-flipping loops in the K3, L85R, nEmrE, and nT28R[1] mutants, this figure reveals a slight improvement in the correlation ($R^2 = 0.88$ vs $R^2 = 0.85$) in comparison to the results in *Figure 7A*.

## Robustness checks for the trajectory termination criteria

Alternative trajectory termination criteria are tested to ensure the robustness of the simulated distribution of multispanning topologies presented in *Figure 2*. As a first alternative, the original set of CG trajectories are extended by 50 s, and the distribution of topologies is determined from the position of the slowest-flipping loop at the end of the extended trajectories. As a second alternative, the distribution of topologies is calculated from the subset of original CG trajectories that reach fully

integrated topologies in which all beads in the TMDs are at least $20\sigma$, rather than $4.5\sigma$, from the origin. These robustness checks are presented in *Figure 2—figure supplement 1* and exhibit excellent correlation with the results obtained using the original protocol.

Additionally, *Figure 7B* shows results in which the CG trajectories are terminated at a range of fixed times following the end of ribosomal translation. Again, the results using this alternative trajectory termination criterion are in good agreement with the results obtained using the original protocol (indicated in dashed lines in *Figure 7B*).

## Cotranslational integration pathways

From the ensemble of CG trajectories, it is possible to examine the pathways by which individual TMDs undergo Sec-facilitated cotranslational integration. In particular, following the definitions of *Cymer et al. (2014)*, it is possible to characterize each cotranslational TMD integration event as corresponding to either the 'channel-sliding', 'interface-sliding', or 'in-out' pathways. Simulation snapshots illustrating the three pathways are shown in *Figure 3—figure supplement 1*.

Each pathway is defined in terms of the series of intermediate states that are visited by the TMD prior to membrane integration. To characterize these intermediate states, the following geometric regions are defined (see *Figure 1—figure supplement 2*). The channel region is defined as that for which $-2\sigma \leq x \leq 2\sigma$ and $-2\sigma \leq y \leq 2\sigma$, the membrane region is defined as that for which $-2\sigma \leq x \leq 2\sigma$ and $y > 2\sigma$, the ribosome region is defined as that for which $-11\sigma \leq x < -2\sigma$ and $-8.5\sigma \leq y \leq 4.5\sigma$, and the cytoplasm region is defined as the region outside of the ribosome for which $x < -2\sigma$. Finally, a bead is considered to overlap the LG if it is within a distance of $\sigma$ to any lateral-gate bead.

We now define the four intermediate states. Intermediate state 1 (IS1) is that for which the TMD partially enters the channel; it is defined as the set of configurations for which at least two TMD beads are in the channel region and zero TMD beads are in the membrane region. Intermediate state 2 (IS2) is that for which the TMD fully spans the membrane while in the channel; it is defined as the set of configurations for which all four TMD beads are in the channel region and the two hydrophilic beads that flank the TMD occupy opposite sides of the membrane. Intermediate state 3 (IS3) is that for which the TMD accesses the membrane interior via the LG; it is defined as the set of configurations for which at least one TMD bead occupies the membrane region, the remaining three TMD beads occupy either the channel or membrane regions, and at least one TMD bead overlaps with the LG. Intermediate state 4 (IS4) is that for which the TMD accesses the cytoplasm region without accessing the channel region; it is defined as the set of configurations for which each of the four TMD beads occupies either the ribosome, membrane, or cytoplasm regions and for which at least one of the hydrophilic beads that flank the TMD occupies the cytoplasm region.

In this analysis, cotranslational TMD integration events are defined as those for which the TMD reaches a membrane integrated configuration (for which all four beads of the TMD span the membrane region and the two hydrophilic flanking beads occupy opposite sides of the membrane) before reaching a misintegrated configuration (for which both hydrophilic flanking beads occupy the same side of the membrane and for which all TMD beads and both flanking beads lie outside of the channel and ribosome regions). Using the definitions of intermediate states, the cotranslational integration pathways are defined as follows. In the 'channel-sliding' pathway, the TMD partially enters the channel, then crosses the LG, then fully integrates into the membrane; a trajectory thus exhibits this pathway if a TMD visits IS1, IS2, and membrane integration in chronological order and without visiting any other intermediate states. In the 'interface-sliding' pathway, the TMD enters the cytoplasm through the gap between the translocon and ribosome, prior to undergoing membrane integration; a trajectory thus exhibits this pathway if a TMD visits IS4 on the way to membrane integration. In the 'in-out' pathway, the TMD fully spans the channel prior to membrane integration; a trajectory thus exhibits this pathway if a TMD visits IS3 on the way to membrane integration without visiting IS4.

At right, *Figure 3—figure supplement 1* shows the relative fraction of cotranslational TMD integration events that exhibit each of these three pathways. It is clear that the dominant cotranslational integration pathway for all four TMDs in both the EmrE and nEmrE mutants is the 'channel-sliding' pathway. This same pathway was also observed in the previous study of single-spanning proteins using the CG model (*Zhang and Miller, 2012a*) and similar configurations were observed in long-timescale atomistic molecular dynamics simulations of the early stages of cotranslational membrane integration (*Zhang and Miller, 2012b*). We find that only a small number

of CG trajectories exhibit the 'interface-sliding' pathway. Finally, we note that the dominant cotranslational integration pathway is likely to depend on the IMP sequence, and the 'channel-sliding' behavior may be less dominant in other IMPs with less hydrophobic TMDs.

## Acknowledgements

Research reported in this publication was supported by the National Institute of General Medical Sciences of the National Institutes of Health under Ruth L Kirschstein National Research Service Award award number 1F32GM113334-01. Computational resources were provided by the National Energy Research Scientific Computing Center, which is supported by the Office of Science of the US Department of Energy under Contract No. DE-AC02-05CH11231, and by the National Science Foundation under Grant No. CHE-1040558.

## Additional information

### Funding

| Funder | Grant reference | Author |
| --- | --- | --- |
| National Institute of General Medical Sciences (NIGMS) | Ruth Kirschtein Postdoctoral Fellowship 1F32GM113334-01 | Reid C Van Lehn |

The funder had no role in study design, data collection and interpretation, or the decision to submit the work for publication.

### Author contributions

RCVL, BZ, TFM, Conception and design, Acquisition of data, Analysis and interpretation of data, Drafting or revising the article

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

## Appendix 1

The CG simulation model used in this work has been previously used to study single-spanning membrane proteins (*Zhang and Miller, 2012a*). Here, the key features of the model are summarized, and the minor modifications introduced in the current work are emphasized.

### Coordinate system

The CG model projects all system coordinates onto a 2D plane passing through the LG of the translocon channel. The coordinate system is defined with the x-axis passing through the channel perpendicular to the bilayer and the y-axis running parallel to the bilayer as shown in *Figure 1—figure supplement 2*. The origin is set at the center of the channel such that negative x-values correspond to cytoplasmic locations. The LG is positioned in the positive y-direction and facilitates membrane entry.

### Interactions

Interactions in the CG model are defined in terms of the reduced units $\epsilon$ and $\sigma$, where $\epsilon = 0.833$ $kT$ sets the energy scale and $\sigma = 0.8$ nm sets the length scale. $\sigma$ is equivalent to the diameter of a CG bead. $T$ is equal to 300 K in all simulations and $k$ is the Boltzmann constant.

The nascent protein is treated as a freely jointed chain with connectivity enforced by a finite extension nonlinear elastic potential between consecutive beads,

$$U_{bond}(r) = -1/2 K R_0^2 \ln\left(1 - r^2/R_0^2\right),$$ (3)

where $r$ is the distance between beads, $K = 7\epsilon/\sigma^2$ is the spring constant and $R_0 = 2\sigma$.

Short-ranged nonbonding interactions are modeled using a cutoff Lennard-Jones (LJ) potential,

$$U_{LJ}(r) = \begin{cases} 4\epsilon_{ij}\left[\left(\frac{\sigma}{r}\right)^{12} - \left(\frac{\sigma}{r}\right)^6\right] + \epsilon_{cr} & r_{cl} < r \le r_{cr} \\ 0 & r \le r_{cl}, r > r_{cr} \end{cases},$$ (4)

where $\epsilon_{ij}$ defines the strength of the interaction between bead $i$ and $j$, $r_{cl}$ is a cutoff at low values of $r$, and $r_{cr}$ is a cutoff at large values of $r$. $\epsilon_{cr}$ is a constant to shift the potential to 0 at $r_{cr}$. The LJ interactions among beads of the nascent protein and between beads of the nascent protein and the ribosome are chosen to be purely repulsive by setting $r_{cr} = 2^{1/6}\sigma$ and $\epsilon_{ij} = \epsilon$. Weak attractive interactions are included between of the nascent protein and beads in the translocon by setting $r_{cr} = 2.5\sigma$ and $\epsilon_{ij} = 1.5\epsilon$. $r_{cl}$ is set to 0 for all interactions between CG beads with the exception of interactions between the nascent protein and the LG in the open state, in which case $r_{cl} = \sigma$ to allow beads to exit the channel and enter the membrane.

Electrostatic interactions between charged beads are modeled using the Debye-Hückel potential,

$$U_{DH}(r) = \begin{cases} \sigma q_i q_j (\beta r)^{-1} \exp(-r/\kappa) & r \ge \sigma \\ q_i q_j \beta^{-1} \exp(-\sigma/\kappa) & r < \sigma \end{cases},$$ (5)

where $\kappa = 1.4\sigma$ approximates the Debye screening length in typical biological media, $\beta = 1/kT$, and $q$ is the charge of a CG bead. The electrostatic potential is capped from below to prevent singularities as beads exit the channel and interact with charges on the open LG.

The effect of the implicit bilayer is modeled by calculating the solvation energy of CG beads,

$$U_{solv}(x, y) = g_i S(x; \phi_x, \psi_x)\left[1 - S(y; \phi_y, \psi_y)\right],$$ (6)

where $g_i$ is the transfer free energy of bead $i$ and $S(x,\phi,\psi)$ is a switching function that defines the membrane region,

$$S(x; \phi, \psi) = \frac{1}{4} \left[ 1 + \tanh\left(\frac{x - \phi}{0.25\sigma}\right) \right] \left[ 1 - \tanh\left(\frac{x - \psi}{0.25\sigma}\right) \right]. \tag{7}$$

This form of the solvation energy and switching function defines the implicit bilayer as the region where $\phi_x < x < \psi_x$ and $y < \psi_y$ or $y > \phi_y$, where $\phi_x = -2.0\sigma$, $\psi_x = 2.0\sigma$, $\phi_y = -1.5\sigma$, and $\psi_y = 1.5\sigma$. The transfer free energy, $g$, for each bead type is approximated from the Wimley–White hydrophobicity scale which measures water–octanol transfer free energies (**Wimley et al., 1996**). Values of $g$ for the different bead types are summarized in **Appendix table 1**. Bead types for all EmrE variants studied in this work are listed in **Appendix table 2**.

**Appendix table 1**. CG bead charges ($q$) and water/membrane transfer free energies ($g$)

|  | Q | L | V | K |
|---|---|---|---|---|
| $q$ | 0.0 | 0.0 | 0.0 | 2.0 |
| $g/\epsilon$ | 2.0 | −4.0 | −2.0 | 6.0 |

**Appendix table 2**. CG bead sequences for each of the 16 EmrE mutants and the cotranslationally-biased mutant

| Protein | L1 | TMD1 | L2 | TMD2 | L3 | TMD3 | L4 | TMD4 | L5 |
|---|---|---|---|---|---|---|---|---|---|
| EmrE | QQQQQ | VVV | KQKQQ | LLLL | QQQQQ | LLLL | KQQQQ | LLLL | KQQQQ |
| EmrE(N$_{cyto}$) | QQQQQ | VVV | KQQQQ | LLLL | QQQQQ | LLLL | QQQQQ | LLLL | KQQQK |
| EmrE(N$_{peri}$) | QQQQQ | VVV | KKKQQ | LLLL | QQQQQ | LLLL | KKQQQ | LLLL | QQQQQ |
| K3 | KQQQQ | VVV | KQKQQ | LLLL | QQQQQ | LLLL | KQQQQ | LLLL | KQQQQ |
| T28R | QQQQQ | VVV | KKKQQ | LLLL | QQQQQ | LLLL | KQQQQ | LLLL | KQQQQ |
| A52K | QQQQQ | VVV | KQKQQ | LLLL | QQKQQ | LLLL | KQQQQ | LLLL | KQQQQ |
| L85R | QQQQQ | VVV | KQKQQ | LLLL | QQQQQ | LLLL | KKQQQ | LLLL | KQQQQ |
| R111 | QQQQQ | VVV | KQKQQ | LLLL | QQQQQ | LLLL | KQQQQ | LLLL | KQQQK |
| nEmrE | QQQQQ | VVV | QQQQQ | LLLL | QQQQQ | LLLL | QQQQQ | LLLL | QQQQQ |
| nK3 | KQQQQ | VVV | QQQQQ | LLLL | QQQQQ | LLLL | QQQQQ | LLLL | QQQQQ |
| nT28R[1] | QQQQQ | VVV | KQQQQ | LLLL | QQQQQ | LLLL | QQQQQ | LLLL | QQQQQ |
| nT28R[2] | QQQQQ | VVV | KQKQQ | LLLL | QQQQQ | LLLL | QQQQQ | LLLL | QQQQQ |
| nT28R | QQQQQ | VVV | KKKQQ | LLLL | QQQQQ | LLLL | QQQQQ | LLLL | QQQQQ |
| nA52K | QQQQQ | VVV | QQQQQ | LLLL | QQKQQ | LLLL | QQQQQ | LLLL | QQQQQ |
| nL85R | QQQQQ | VVV | QQQQQ | LLLL | QQQQQ | LLLL | KKQQQ | LLLL | QQQQQ |
| nR111 | QQQQQ | VVV | QQQQQ | LLLL | QQQQQ | LLLL | QQQQQ | LLLL | KQQQQ |

| CB mutant | L1 | TMD1 | L2 | TMD2 | L3 | TMD3 | L4 | TMD4 | L5 |
|---|---|---|---|---|---|---|---|---|---|
|  | KQQQK | LLLL | QQQQQ QQQQQ | LLLL | KQQQQ QQQQK | LLLL | QQQQQ QQQQQ | LLLL | KQQQQ QQQQK |

Moderately hydrophobic (V), very hydrophobic (L), neutral-hydrophilic (Q) and charged (K) beads are colored as light gray, gray, black, and red respectively. Loops and TMDs are labeled.

## System dynamics

The time-evolution of the nascent protein is modeled with Brownian dynamics with the equation of motion,

$$x_i(t + \Delta t) = x_i(t) - \beta D \frac{\partial V(\mathbf{x}(t))}{\partial x_i} \Delta t + \sqrt{2D\Delta t}\eta_i, \tag{8}$$

where $x(t)_i$ is the position (in two dimensions) of bead $i$ at time $t$, $V(\mathbf{x}(t))$ is the potential energy of the system, $D$ is the isotropic bead diffusion coefficient, $\Delta t$ is the time step, and $\eta_i$ is random

noise drawn from a Gaussian distribution with zero mean and a variance of one. Only beads in the nascent protein are subject to Brownian dynamics. $D$ is set to a value of 758.7 nm²/s and $\Delta t$ is set to 100 ns.

LG dynamics are modeled by attempting to stochastically open or close the LG at every simulation timestep with a probability $p_{open} = k_{open}\Delta t$ or $p_{close} = k_{close}\Delta t$, where

$$k_{open} = \frac{1}{\tau_{LG}} \frac{\exp(-\beta\Delta G_{tot})}{1 + \exp(-\beta\Delta G_{tot})}, \tag{9}$$

$$k_{close} = \frac{1}{\tau_{LG}} \frac{1}{1 + \exp(-\beta\Delta G_{tot})}. \tag{10}$$

$\tau_{LG} = 500$ ns is the timescale for attempting LG opening/closing and $\Delta G_{tot}$ is the change in the free energy for opening the LG. $\Delta G_{tot}$ depends on the presence of the nascent protein beads in the channel and is defined as

$$\Delta G_{tot} = \sum_{i=1}^{M} g_i + \Delta E + \Delta G_{empty}\chi_{empty}, \tag{11}$$

where $M$ is the number of beads occupying the translocon, $\Delta E$ is the difference between the total LG/protein LJ interactions in the closed state and total LG/protein LJ interactions in the open state, $\Delta G_{empty} = 16\epsilon$ is the free energy cost for opening the LG when there is no nascent protein in the channel, and $\chi_{empty}$ is the fraction of the channel that is empty for a given timestep. The first term promotes LG opening when hydrophobic beads enter the channel, the second term prevents the LG from closing when occluded by a nascent protein, and the third term promotes LG closing once the nascent protein exits the channel. Additional details on the development and numerical testing of the CG model are provided in **Zhang and Miller (2012a)**.

## Wall potential

A modification that appears in the current implementation of the CG model is a wall potential that prevents the nascent protein from returning to the translocon once it completes translation and diffuses a given distance away from the channel. The potential has the form

$$U_{wall}(y) = \begin{cases} 1/2K_{wall}(y - 10\sigma)^2 & y < 10\sigma \\ 0 & y \geq 10\sigma \end{cases}, \tag{12}$$

where the spring constant, $K_{wall}$, is set to 10 $\epsilon/\sigma^2$. The potential is only added to the system when all beads of the nascent protein have y-positions greater than $12\sigma$. Inclusion of the wall potential was found to avoid artifacts associated with the nascent protein interacting with the translocon long after exiting the channel. These artifacts were expected to be accentuated in the CG model due to its reduced dimensionality; nonetheless, the results are qualitatively unchanged if the wall potential is not included.

