## [Decision Letter]

Thank you for submitting your work entitled “Regulation of multispanning membrane protein topology via post-translational annealing” for peer review at *eLife*. Your submission has been favorably evaluated by Randy Schekman (Senior Editor) and three reviewers, one of whom, Gerhard Hummer, is a member of our Board of Reviewing Editors.

he reviewers have discussed the reviews with one another and the Reviewing Editor has drafted this decision to help you prepare a revised submission.

Co-translational and post-translational steps in Sec-facilitated insertion of the EmrE integral membrane protein are studied using a coarse-grained simulation model. Consistent with experiment, the simulations produce a mixture of orientations of the protein in the membrane that is sensitive to mutations in the loops connecting the transmembrane helices. Transient topologies are found to anneal by flipping loops across the membrane during post-translational annealing. The resulting orientation distribution thus depends strongly on the electrostatic character of these loops. This work sheds light on the mechanism of membrane protein insertion, its kinetics, and the factors determining the orientation and its variability.

All three referees are overall positive but raise a number of concerns regarding the details of the model, the interpretation of the simulation results, and the presentation in the paper. I find the issues raised by the referees to be relevant. Addressing them in a revision should result in a stronger paper. Following is a brief summary of their reports that emphasizes the main points as seen from my perspective:

Reviewer #1 is concerned that the outcome of the simulations was sensitive to the somewhat arbitrary simulation termination criterion. Other concerns center on the modeling of the peptides, including the assignment of CG sequences, loop length, and the neglect of negative charge (however, the latter may not be particularly relevant for the EmrE sequence).

Reviewer #2 points out a number of relevant connections to the literature, and asks for clarifications/rectifications in the presentation of the model and the simulation results.

Reviewer #3 is concerned that the reported agreement with experiment, while visually strong, may not be quite as convincing in a more quantitative statistical analysis. The reviewer is also concerned about the factors determining the rates of loop flipping in the model and in real systems. The role of degradation (which could affect slowly relaxing EOT structures) should also be discussed in more detail.

In the following, I highlight the points in the reports I consider most relevant.

1) The terms topology, single-topology, and dual-topology should be explained to the general reader. The exact meaning of these technical terms is not immediately obvious and has caused some confusion even among the experts (cf. dual vs. antiparallel topology).

2) The simulations are terminated when a dissociated N_cyto_/C_cyto_ or N_peri_/C_peri_ state with 4 inserted TMDs is reached. The authors should briefly discuss (a) if the results are sensitive to variations in the termination criterion (e.g. by continuing the simulations for an additional set time, by asking for a larger distance from the origin, by running the simulations for a fixed time after initial release etc.) and (b) if there is a justification for this criterion (e.g. that subsequent changes to the topology occur on much longer time scales). It would also be useful to indicate the length of the longest simulation at which 0% of misintegrated proteins is observed.

3) Proteins with mixed orientations in the membrane appear to be the exception. The authors should thus briefly discuss whether by applying the same coarse-graining recipe to other proteins, and using the same simulation protocol, one indeed finds well defined orientations.

4) TMD1 was treated differently from the other 3 TMDs. What happens if one uses the same bead type for all TMDs? (The text states that L-type beads were used, but Supplementary file 1C lists them as V-type beads. The legend of the table also has the assignment of moderate and strong hydrophobicity of L and V beads switched vis-à-vis the text).

5) A discussion of the present results in light of earlier studies should prove insightful. In particular, Cymer et al. in a recent review (J Mol. Biol., 427:999-1022, 2015) raised questions about the simplistic model of release of TM helices in the correct topology. They noted that (a) the assembly process is driven by thermodynamics and (b) at an elongation rate of about 50 ms per amino acid, there is ample time for a chain to sample what could be a very complicated energy landscape. Here the authors indeed find that the insertion pathway and topology of a given EmrE mutant are not immutable. This connection should be discussed.

6) The pathways of insertion should be discussed in more detail. Figure 1 shows the 2D model used by Miller et al. The nascent chain can exit from the translocon either by a gap between translocon and membrane or through the translocon. Although the upper panel of Figure 1 suggests passage through the translocon, the lower panel shows the nascent chain passing from ribosome directly into the membrane. The paper does not address which of these two pathways is preferred. I think the authors should comment on this and, if possible, provide more information on the pathways observed in the simulations. This is important, because Cymer et al. (2015) suggested that the preferred path of the nascent chain might be along the translocon lateral gate rather than through the translocon as suggested by the current cartoons. The lower panel of Figure 1 suggests that the lateral gate pathway may be the preferred path.

7) A more thorough statistical analysis of the data would strengthen the paper. P-values, rank correlations, or other measures of correlation could be calculated for the data presented in Figure 2 so that the reader can understand the correlation and statistical significance of the simulation results compared to the experimental results.

8) What is the role of neighboring protein elements on the flipping frequency of a given loop? How do these correlated motions affect the flipping frequencies presented in Figure 6?

9) A more detailed discussion of the possible role of degradation would be important. Based on experimentally determined timescales of degradation (a brief comment is presented in the last paragraph of the Discussion), and the time scales of insertion and annealing observed here, can one estimate the degraded fraction and the effect on the orientation distribution?

Along the same lines, in the third paragraph of the subsection “Position of slowest-flipping loop in EOT ensemble determines fully integrated topology”, it is not immediately clear how “the fraction of CG trajectories that have reached fully integrated multispanning configurations” (Figure 7) is related to degradation of sequences as a function of time prior to reaching a fully integrated multispanning topology. This paragraph needs to be edited for greater clarity to reveal the authors' assumptions more clearly.

[Editors' note: further revisions were requested prior to acceptance, as described below.]

Thank you for resubmitting your work entitled “Regulation of multispanning membrane protein topology via post-translational annealing” for further consideration at *eLife*. Your revised article has been favorably evaluated by Randy Schekman (Senior Editor), a Reviewing Editor, and two reviewers. The manuscript has been improved but there are some remaining issues that need to be addressed before acceptance, as outlined below:

Please discuss the recent study of Woodall et al. (Nat. Comms, 6:8099; DOI: 10.1038/ncomms9099) on EmrE topologies.

Following are the full reports of the reviewers.

Reviewer #2:

The authors have, in my opinion, responded extremely well to the previous reviews. Critical questions raised have been addressed by the performance of additional simulations and analyses. The conclusions of the paper have been strengthened significantly as a result.

A new issue has arisen, however. Woodall et al. (Nat. Comms 6:8099; DOI: 10.1038/ncomms9099) have just reported (26 August) experiments directed at answering the question of how a cell can generate equal populations of two opposite topologies of EmrE. An important conclusion is that the FtsH protease of *E. coli* cleans up misfolded or unpaired EmrEs to create equal populations of both EmrE topologies. The authors should address these new findings, particularly in the context of their Figure 7.

Reviewer #3:

The authors have satisfactorily addressed my concerns. I recommend the manuscript's publication.

---

## [Author Response]

1) The terms topology, single-topology, and dual-topology should be explained to the general reader. The exact meaning of these technical terms is not immediately obvious and has caused some confusion even among the experts (cf. dual vs. antiparallel topology).

We agree that clarification and unambiguous usage of these terms is essential.

Throughout the manuscript, we use “topology” to indicate the orientation of a fully integrated, multispanning protein with respect to the cell membrane. A given IMP can thus become fully integrated in one of two topologies (which for EmrE correspond to N_peri_/C_peri_ and N_cyto_/C_cyto_). To emphasize that “topology” refers to the fully integrated proteins, as opposed to partially integrated configurations of the proteins, we at times refer to the “fully integrated topology” – however, in both cases, we are referring to the same notion and the more verbose term is used to avoid confusion.

“Dual-topology proteins” refers to those which exhibit both possible orientations of the fully integrated IMP with respect to the cell membrane in approximately 1:1 stoichiometry (Rapp et al., Nat. Struct. Biol., 2006; Rapp et al., Science, 2007).

The term “single-topology” has been eliminated from the manuscript, as it is not needed and only creates potential confusion.

The term “antiparallel” is used only in reference to dimers of IMPs with opposite topologies.

To address this comment in the revised manuscript, the first two paragraphs of the Introduction have been modified to clarify the usage of “topology” and “dual-topology”. The text in the subsection “Alternative definition of the slowest-flipping loop position for mutants with two slow-flipping loops” has been modified to clarify the usage of “antiparallel.” We have eliminated usage of the phrase “overall topology” throughout the text to avoid confusion.

*2) The simulations are terminated when a dissociated N*_*cyto*_*/C*_*cyto*_
*or N*_*peri*_*/C*_*peri*_
*state with 4 inserted TMDs is reached. The authors should briefly discuss (a) if the results are sensitive to variations in the termination criterion (e.g. by continuing the simulations for an additional set time, by asking for a larger distance from the origin, by running the simulations for a fixed time after initial release etc.) and (b) if there is a justification for this criterion (e.g. that subsequent changes to the topology occur on much longer time scales). It would also be useful to indicate the length of the longest simulation at which 0% of misintegrated proteins is observed.*

With reference to point (a), we find that the distribution of topologies presented in Figure 2 is not sensitive to variations in the trajectory termination criteria. Per the recommendations of the referees, we tested (i) the effect of extending each original CG trajectory by 50 s and (ii) the effect of increasing the distance from the origin that was used to define the trajectory cutoff in the original protocol. These new robustness checks are presented in Figure 2—figure supplement 1 and show no significant deviations from the original results in Figure 2. Similarly, the results in Figure 7 show that terminating simulations at a set time after the end of translation does not affect the distribution of topologies in Figure 2.

To address this point in the manuscript, details of the new calculations have been added to the new section “Robustness checks for the trajectory termination criteria” in the Materials and methods.

To address point (b) in the manuscript, we provide biophysical justification for the trajectory termination criteria, adding the following explanation in the subsection “Simulation protocol”:

“The trajectory termination criteria are designed to examine the effects of the Sec-facilitated membrane integration process on EmrE topogenesis. […] Demonstration of the robustness of the reported results to the cutoff values employed in the trajectory termination criteria are provided in the Robustness checks for the trajectory termination criteria section of the Materials and methods.”

The choice of trajectory termination criteria is further justified by the robustness of the reported results with respect to the detailed cutoff values, as discussed with regard to point (a).

Finally, in regard to the reviewers’ suggestion to report the longest simulation at which 0% of misintegrated proteins is observed, we have included a table in Figure 3—figure supplement 2 that reports the amount of simulation time required for 50%, 90%, and 95% of the CG trajectories to reach fully integrated topologies for each mutant. This table provides the reader with a useful sense of the range of lengths of the trajectories. (We prefer not to report the “longest trajectory”, since this is not a well-defined property of the converged ensemble and will become arbitrarily long as the number of trajectories in the ensemble of simulated trajectories increases.)

3) Proteins with mixed orientations in the membrane appear to be the exception. The authors should thus briefly discuss whether by applying the same coarse-graining recipe to other proteins, and using the same simulation protocol, one indeed finds well defined orientations.

This is a good point. The coarse-grained model has indeed been tested for the prediction of topology in membrane proteins with a single dominant topology.

In previous work (B. Zhang and T. F. Miller III, Cell Rep., 2012), the topogenesis of single-spanning signal anchor proteins was shown to exhibit similar trends as in the experimental studies of Goder and Spiess (EMBO J., 2003). It was further shown in this previous work that a multispanning (three TMD) membrane protein with a strong positive charge bias obtained the correct, single dominant topology according to the positive-inside rule, in agreement with expectations.

In the current work, the EmrE, EmrE(N_cyto_), and EmrE(N_peri_) mutants, which have topologies that have been thoroughly characterized experimentally (Rapp et al., Science, 2007; Seppala et al., Science, 2010), exhibit the expected distribution of topologies. The cotranslationally-biased mutant provides another example where incorporating a strong positive charge bias leads to a single dominant topology as expected from the positive-inside rule. These examples underscore the ability of the CG model to produce well-defined topologies.

To emphasize past findings in the revised manuscript, the last paragraph of the subsection “Coarse-grained model” has been expanded to read: “Moreover, the model has been shown to accurately predict features of single-spanning IMP integration and topogenesis […] The strong agreement between simulation and experimental results presented in this work further indicates that IMP topological determinants are captured at this CG resolution.”

4) TMD1 was treated differently from the other 3 TMDs. What happens if one uses the same bead type for all TMDs? (The text states that L-type beads were used, but Supplementary file 1C lists them as V-type beads. The legend of the table also has the assignment of moderate and strong hydrophobicity of L and V beads switched vis-à-vis the text).

The representation for TMD1 used in the model is based on the hydropathy plot presented in Figure 1—figure supplement 1, which shows that the first TMD is significantly less hydrophobic than the following three.

Regardless, we agree that it is worth investigating whether this lower hydrophobicity of TMD1 impacts the reported conclusions; having performed additional simulations to investigate this point, we find that it does not have a significant impact.

To test the robustness of our results with respect to TMD1 hydrophobicity, Figure 5—figure supplement 1 presents the end-of-translation (EOT) ensemble averaged loop positions for the six EmrE mutants. The loop positions are compared between mutants with either a moderately hydrophobic (V-type beads) or very hydrophobic (L-type beads) TMD1. The positions of loops L3-L5 are largely unchanged in all cases. There is a slight increase in L2 cytoplasmic retention for all six mutants; this finding agrees with the increase in type I (N_peri_/C_cyto_) integration with increasing TMD hydrophobicity that was identified in previous work (B. Zhang and T. F. Miller III, Cell Rep., 2012). Given the strong correlation (Figure 7) between EOT loop positions and the final, fully integrated topology, these results suggest that increasing TMD1 hydrophobicity does not qualitatively affect the reported conclusions.

We have revised the manuscript to include this additional robustness check as follows. We have added Figure 5—figure supplement 1 as described above. Furthermore, in the subsection “EmrE protein”, we have added: “Similarly, the results of the simulations are robust with respect to changes in the modeling of TMD1 hydrophobicity (Figure 5—figure supplement 1) and loop length (Figure 3—figure supplement 3).”

As per the reviewers’ suggestion, we have also rechecked the manuscript and figures to confirm that the reported bead-types for TMD1 are correct.

5) A discussion of the present results in light of earlier studies should prove insightful. In particular, Cymer et al. in a recent review (J Mol. Biol., 427:999-1022, 2015) raised questions about the simplistic model of release of TM helices in the correct topology. They noted that (a) the assembly process is driven by thermodynamics and (b) at an elongation rate of about 50 ms per amino acid, there is ample time for a chain to sample what could be a very complicated energy landscape. Here the authors indeed find that the insertion pathway and topology of a given EmrE mutant are not immutable. This connection should be discussed.

We agree that this review raises interesting questions that should be cited in the manuscript.

As the reviewers point out, the translation process is slow, and equilibrium thermodynamic effects play a major role in determining the insertion pathway of a nascent chain, which is also noted in the review by Cymer et al. Nonetheless, some experiments, such as the work of Goder and Speiss (EMBO J., 2003), also demonstrate a clear role for kinetic effects, in addition to purely equilibrium effects, on nascent chain topogenesis.

The CG model has previously been demonstrated (B. Zhang and T. F. Miller III, Cell Rep., 2012) to correctly describe membrane integration processes that are governed by both thermodynamic and/or kinetic effects. In particular, the CG model correctly reproduces the stop-transfer efficiency of TMDs with varying hydrophobicity (T. Hessa et al., Nature, 2005), a process that is dominated by thermodynamics. The CG model also correctly reproduces the effect of translation rate and nascent chain length on signal anchor topogenesis (Goder and Spiess, EMBO J., 2003), both of which exhibit clear kinetic effects. The CG model is therefore able to model pathways dominated by both kinetics and thermodynamics as discussed in the review by Cymer et al.

To emphasize these points in the revised text, we have expanded the last paragraph of the subsection “Coarse-grained model” to read: “Moreover, the model has been shown to accurately predict features of single-spanning IMP integration and topogenesis […] and it has provided a means of understanding the competition between such effects.” Note that this revision includes explicit citation of the Cymer et al. review (2014).

*6) The pathways of insertion should be discussed in more detail.*
Figure 1
*shows the 2D model used by Miller et al. The nascent chain can exit from the translocon either by a gap between translocon and membrane or through the translocon. Although the upper panel of*
Figure 1
*suggests passage through the translocon, the lower panel shows the nascent chain passing from ribosome directly into the membrane. The paper does not address which of these two pathways is preferred. I think the authors should comment on this and, if possible, provide more information on the pathways observed in the simulations. This is important, because Cymer et al. (2015) suggested that the preferred path of the nascent chain might be along the translocon lateral gate rather than through the translocon as suggested by the current cartoons. The lower panel of*
Figure 1
*suggests that the lateral gate pathway may be the preferred path.*

As suggested by the reviewers, we have performed additional analysis of the simulation data to characterize the cotranslational integration pathways. We find that the preferred pathway for cotranslational integration is for a TMD to partially enter the channel, then integrate into the membrane via the lateral gate.

Our analysis of the simulated trajectories examines three pathways by which by which individual TMDs undergo Sec-facilitated cotranslational integration. These three pathways are categorized according to the definitions of Cymer et al. In the “channel-sliding” pathway, the TMD partially enters the channel, then crosses the lateral gate, then fully integrates into the membrane. In the “interface-sliding” pathway, the TMD enters the cytoplasm through the gap between the translocon and ribosome, prior to undergoing membrane integration. In the “in-out” pathway, the TMD fully spans the channel prior to membrane integration.

Figure 3—figure supplement 1 in the revised manuscript shows that the dominant cotranslational integration pathway for all four TMDs in both the EmrE and nEmrE mutants is the “channel-sliding” pathway. This same pathway was also observed in our previous study of single-spanning proteins using the CG model (B. Zhang and T. F. Miller III, Cell Rep., 2012) and similar configurations were observed in previous atomistic molecular dynamics simulations (B. Zhang and T. F. Miller III, JACS, 2012). These observations agree in part with the proposal by Cymer et al. that TMDs do not need to fully enter the channel prior to lateral gate opening and membrane integration. However, we observe only a few CG trajectories in which TMDs exhibit the “interface-sliding” pathway. Finally, we note that the “channel-sliding” behavior may be less dominant in other IMPs with less hydrophobic TMDs.

To provide this analysis of the cotranslational integration pathways in the revised manuscript, we have added Figure 3—figure supplement 1 and a new section titled “Cotranslational integration pathways” to the Materials and methods section. We have also updated trajectory snapshots in Figure 1 to depict the dominant integration pathway and added to the subsection “Dual-topology proteins exhibit slow post-translational integration”: “Over 90% of the CB mutant trajectories reach the N_cyto_/C_cyto_ topology within three seconds following the completion of translation and thus rapidly integrate as expected for the cotranslational model (4; 50; 39); mechanistic features of individual TMD integration steps are discussed in the Cotranslational integration pathways section of the Materials and methods.”

*7) A more thorough statistical analysis of the data would strengthen the paper. P-values, rank correlations, or other measures of correlation could be calculated for the data presented in*
Figure 2
*so that the reader can understand the correlation and statistical significance of the simulation results compared to the experimental results.*

We agree with this comment and have revised the manuscript to include improved statistical analysis.

First, for the data in Figure 2, we calculated the Pearson correlation coefficient to quantify the linear correlation between the simulation and experimental data. This calculation yielded a value of 0.92, confirming a strong linear correlation. We have also graphically illustrated this correlation in Figure 2—figure supplement 2 of the revised text. This plot is divided into quadrants – data points lying in the top right and bottom left quadrants (shaded in light gray) show where simulations and experiments both predict the same dominant topology. All mutants, with the exception of A52K, have the same dominant topology in the simulations as in the experiments within the statistical error. These measures confirm the agreement between simulation/experiment discussed in the text and strengthen the conclusions in the manuscript.

In the revised text, we have added Figure 2—figure supplement 2 and the following passage to the subsection “Simulations match experimental observations of topology”: “Figure 2—figure supplement 2 illustrates that the distribution of topologies determined experimentally and the distribution of topologies measured from the simulations are linearly correlated (Pearson correlation coefficient, *r* = 0.92); points lying in the two shaded quadrants of the graph correspond to proteins for which the simulations and experiments predict consistent topologies.”

*8) What is the role of neighboring protein elements on the flipping frequency of a given loop? How do these correlated motions affect the flipping frequencies presented in*
Figure 6*?*

As suggested by the reviewers, we have performed additional analysis of the CG trajectories to address the role of neighboring protein elements and correlated loop motions on the loop-flipping frequencies.

Our new analysis indicates that multiple loops do not flip concurrently within the same time interval, although loops are more likely to flip when neighboring TMDs are misintegrated. These results show that while loop-flipping events are not concerted on the millisecond timescale, the loop-flipping frequency of a given loop is impacted by the orientation of its neighboring TMDs.

To examine the concurrence of loop-flipping events, we calculated the frequency with which multiple loops undergo flipping within the same 1 ms interval. It was found that pairs of loops undergo concurrent flipping with a frequency that is only 0.015% of the frequency of single loop-flipping events. Three or more loops were never observed to concurrently flip. These results indicate that loop-flipping events are not concerted on the millisecond timescale.

To examine the effect of neighboring TMDs on the loop-flipping frequency, we separately calculate the loop-flipping frequency for each loop for configurations in which zero, one, or two of the neighboring TMDs is misintegrated. On average, it was found that a loop with a single misintegrated neighboring TMD flips 1.5 times more frequently than the same loop with zero misintegrated neighboring TMDs, while a loop with two misintegrated neighboring TMDs flips 3.7 times more frequently than the same loop with zero misintegrated neighboring TMDs. These findings show that the orientation of neighboring TMDs does indeed impact the loop-flipping frequency.

To address these points in the revised manuscript, we have added to the subsection “Calculation of loop-flipping frequency”: “Loop-flipping events are not found to be strongly concerted, as two or more loops were observed to flip concurrently in only 0.015% of all 1-ms time intervals in which at least one loop-flipping event was observed. […] Additional details on these calculations are presented in the Calculation of loop-flipping frequency section of the Materials and methods.” We have also added additional details to the “Calculation of flipping frequency” section.

9) A more detailed discussion of the possible role of degradation would be important. Based on experimentally determined timescales of degradation (a brief comment is presented in the last paragraph of the Discussion), and the time scales of insertion and annealing observed here, can one estimate the degraded fraction and the effect on the orientation distribution?

*Along the same lines, in the third paragraph of the subsection “Position of slowest-flipping loop in EOT ensemble determines fully integrated topology”, it is not immediately clear how “the fraction of CG trajectories that have reached fully integrated multispanning configurations” (*Figure 7*) is related to degradation of sequences as a function of time prior to reaching a fully integrated multispanning topology. This paragraph needs to be edited for greater clarity to reveal the authors' assumptions more clearly.*

We agree that this issue of protein degradation in this context is interesting and worth clarifying. While the exact pathway for EmrE degradation is unknown, several bacterial proteases that degrade membrane proteins have been characterized which provides insight into the approximate degradation timescale (12). Notably, FtsH is a membrane-embedded protease that degrades misassembled IMPs over timescales ranging from 2 minutes (for SecY) to 15 minutes (for YccA) in *E. coli* (K. Ito and Y. Akiyama, Annu. Rev. Microbiol., 2005)*.* In comparison to the simulated trajectories (Figure 3—figure supplement 2), these degradation timescales are relatively slow, supporting the assumption that IMP integration and post-translational annealing reaches completion prior to significant degradation. Nonetheless, if it is assumed that misfolded proteins are uniformly subject to degradation, even on relatively short (i.e. 5-100 s) timescales, we provide analysis in Figure 7 to suggest that degradation still does not significantly affect the distribution of topologies predicted from the simulations.

Our analysis of the likely effects of degradation are presented in Figure 7. This figure presents the relative fraction of N_cyto_/C_cyto_ and N_peri_/C_peri_ protein topologies for the CG trajectories that have reached fully integrated multispanning topologies as a function of time, excluding all trajectories for which at least one TMD is misintegrated If it is assumed that fully integrated proteins are resistant to degradation, then each point in Figure 7 represents the distribution of topologies that would be observed if all misfolded proteins were uniformly degraded at the corresponding time. The figure shows that the distribution of topologies is nearly constant with respect to degradation time, preserving the correlation between the position of the slowest-flipping loop at the end of translation and in the fully integrated multispanning topology.

To clarify these points in the manuscript, we have expanded the last paragraph of the Results.

[Editors' note: further revisions were requested prior to acceptance, as described below.]

Reviewer #2:

The authors have, in my opinion, responded extremely well to the previous reviews. Critical questions raised have been addressed by the performance of additional simulations and analyses. The conclusions of the paper have been strengthened significantly as a result.

*A new issue has arisen, however. Woodall et al. (Nat. Comms 6:8099; DOI: 10.1038/ncomms9099) have just reported (26 August) experiments directed at answering the question of how a cell can generate equal populations of two opposite topologies of EmrE. An important conclusion is that the FtsH protease of* E. coli *cleans up misfolded or unpaired EmrEs to create equal populations of both EmrE topologies. The authors should address these new findings, particularly in the context of their*
Figure 7*.*

We are pleased that all previous issues raised by this reviewer have been addressed in the revised manuscript. We agree that the new paper referenced by the reviewer merits discussion in the context of our work, and the manuscript has been revised to introduce several points of contact with the study, particularly in regard to the FtsH degradation results discussed in Figure 7.

We have modified the sentence in the third paragraph of the subsection “Simulation protocol” of the revised manuscript to read: “Specifically, it is assumed that upon reaching configurations in which all of the TMDs are integrated into the membrane, the protein topology remains irreversibly fixed for all subsequent times; physical processes that may lead to this irreversibility include the dimerization of EmrE proteins to form functional channels in the membrane (31) or the degradation of undimerized EmrE proteins prior to topological inversion (53).”

We have also modified the subsection “Position of slowest-flipping loop in EOT ensemble determines fully integrated topology” to read: “Several bacterial proteases that degrade membrane proteins have been characterized which provides insight into the approximate degradation timescale […] then each point in Figure 7 represents the distribution of topologies that would be observed if all misfolded proteins were ussniformly degraded at the corresponding time.”

Finally, we have added to the first paragraph of the Discussion: “The proposed mechanism also agrees with recent experiments that find EmrE to undergo partial topological rearrangements that correspond to the loop-flipping events described here (Woodall, Yin and Bowie, 2015).”